# Wikipedia in the Era of LLMs: Evolution and Risks

**Siming Huang**[†]
*Huazhong University of Science and Technology*

**Yuliang Xu**[†]
*Huazhong University of Science and Technology*

**Mingmeng Geng**[*]  *mingmeng.geng@ens.psl.eu*
*École Normale Supérieure (ENS) – Université Paris Sciences et Lettres (PSL) & Laboratoire LATTICE*
*International School for Advanced Studies (SISSA)*

**Yao Wan**[*]  *wanyao@hust.edu.cn*
*Huazhong University of Science and Technology*

**Dongping Chen**[‡]
*Huazhong University of Science and Technology*

**Reviewed on OpenReview:** *https://openreview.net/forum?id=ahVmnYkVLt*

## Abstract

In this paper, we present a comprehensive analysis and monitoring framework for the impact of Large Language Models (LLMs) on Wikipedia, examining the evolution of Wikipedia through existing data and using simulations to explore potential risks. We begin by analyzing article content and page views to study the recent changes in Wikipedia and assess the impact of LLMs. Subsequently, we evaluate how LLMs affect various Natural Language Processing (NLP) tasks related to Wikipedia, including machine translation and retrieval-augmented generation (RAG). Our findings and simulation results reveal that Wikipedia articles have been affected by LLMs, with an impact of approximately 1% in certain categories. If the machine translation benchmark based on Wikipedia is influenced by LLMs, the scores of the models may become inflated, and the comparative results among models could shift. Moreover, the effectiveness of RAG might decrease if the knowledge has been contaminated by LLMs. While LLMs have not yet fully changed Wikipedia's language and knowledge structures, we believe that our empirical findings signal the need for careful consideration of potential future risks in NLP research.[1]

## 1 Introduction

The creation of Wikipedia challenged traditional encyclopedias (Giles, 2005), and the rapid development and wide adoption of Large Language Models (LLMs) have sparked concerns about the future of Wikipedia (Wagner & Jiang, 2025; Vetter et al., 2025). Researchers have begun examining the influence of LLMs on Wikipedia, and it is unlikely that Wikipedia has remained unaffected. For example, Reeves et al. (2024) analyze Wikipedia user metrics such as page views and edit histories. Meanwhile, Brooks et al. (2024) estimate the proportion of AI-generated content in newly created English Wikipedia articles using Machine-Generated Text (MGT) detectors. Given the richness and significance of Wikipedia, the impact of LLMs on Wikipedia requires a more comprehensive and detailed investigation.

Wikipedia is widely recognized as a valuable resource (Singer et al., 2017), and its content is extensively utilized in AI research, particularly in Natural Language Processing (NLP) tasks (Johnson et al., 2024b).

---

† Equal Contribution. * Corresponding Authors. ‡ Project Lead.
[1]We release all the experimental dataset and source code at: https://github.com/HSM316/LLM_Wikipedia.

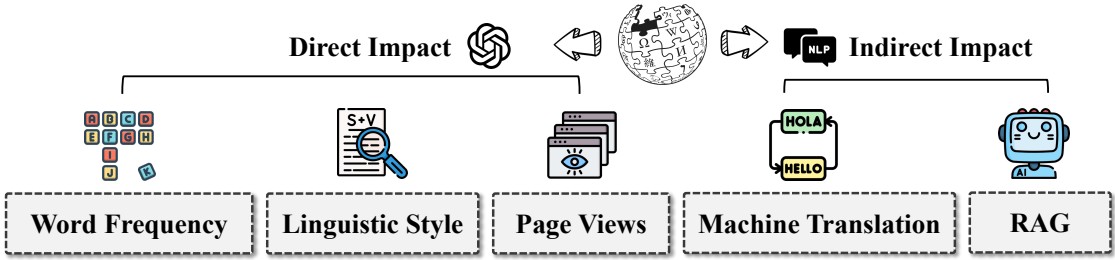

Figure 1: Our work analyzes the direct impact of LLMs on Wikipedia, and explores the indirect impact of LLM-generated content on Wikipedia: **Have LLMs already impacted Wikipedia, and if so, how might they influence the NLP community and human society?**

For instance, Wikipedia pages are among the five datasets used to train GPT-3 (Brown et al., 2020). The sentences in the *Flores-101* evaluation benchmark are extracted from English Wikipedia (Goyal et al., 2022). Lewis et al. (2020)'s work on Retrieval-Augmented Generation (RAG) treated Wikipedia as a source of factual knowledge. Therefore, we aim to investigate the influence of LLMs on machine translation and knowledge systems using Wikipedia as a key resource.

Figure 1 illustrates the various tasks and research topics discussed in this paper. Our first objective is to evaluate the direct impact of LLMs on Wikipedia, focusing on *word frequency*, *linguistic style*, and *page views*. Then we explore the indirect effects on the broader NLP community, particularly in relation to *machine translation benchmarks* and *RAG*, both of which rely heavily on Wikipedia content for their corpora. This positions us to better observe and assess the evolutions and risks of Wikipedia in the era of LLMs. Our analysis yields a number of significant insights:

- Some Wikipedia pages have been influenced by LLMs, although the overall impact remains limited so far.

- There has been a slight decline in page views for certain scientific categories on Wikipedia, and similar trends are observed across multiple language editions, although the connection to LLMs remains uncertain.

- If the sentences in machine translation benchmarks are drawn from LLM-influenced Wikipedia content, the evaluated models' scores may be inflated, potentially reversing their relative rankings.

- Wikipedia content processed by LLMs could be less effective for RAG compared to real Wikipedia content.

Based on these findings, we underscore the importance of carefully assessing potential risks and encourage further exploration of these issues. The key contributions of this paper are three-fold, as we are the first to: (1) quantify the impact of LLMs on Wikipedia pages across various categories; (2) analyze the impact of LLMs on Wikipedia based on word usage and provide the corresponding estimates; and (3) examine how LLM-generated content affects machine translation evaluation and the efficiency of RAG systems.

This is also very likely the first paper to comprehensively analyze the impact of LLMs on Wikipedia based on data and simulations. It is important to note that, while some changes are not obvious at the moment, the methods and perspectives we have proposed can be employed for long-term detection of the impact of LLMs on Wikipedia in the future.

## 2 Related Work

**Wikipedia for NLP.** Wikipedia has long been utilized in various NLP applications (Strube & Ponzetto, 2006; Mihalcea & Csomai, 2007; Zesch et al., 2008; Gabrilovich & Markovitch, 2009; Navigli & Ponzetto, 2010). In the era of LLMs, Wikipedia also plays an important role, such as in fact-checking (Hou et al., 2024) and reducing hallucinations (Semnani et al., 2023). Writing Wikipedia-like articles is also one of the LLM applications (Shao et al., 2024).

**LLMs for Wikipedia.** Researchers are trying to use LLMs to enhance Wikipedia, including articles (Adak et al., 2025), Wikidata (Peng et al., 2024; Mihindukulasooriya et al., 2024) and edit process (Johnson et al., 2024a). Some researchers have compared LLM-generated or rewritten Wikipedia articles with human-written ones, yielding differing conclusions (Skarlinski et al., 2024; Zhang et al., 2025a).

**Estimation of LLM Impact.** There are studies on the impact of LLMs on its page views (Reeves et al., 2024; Lyu et al., 2025). The detection of AI-generated content has been a hot research topic in recent years (Wu et al., 2025; Wang et al., 2025; Zhang et al., 2024), including its application to Wikipedia articles (Brooks et al., 2024). But MGT detectors have notable limitations (Doughman et al., 2024), and researchers are also exploring other methods for estimating the LLM impact, such as word frequency analysis (Liang et al., 2024; Geng & Trotta, 2024). Moreover, the emergence of LLM-assisted edits on Wikipedia has raised concerns about preserving the encyclopedia's style and editorial standards (Ashkinaze et al., 2024). Contamination of its articles with LLM-generated text can create harmful feedback loops in model training (Shumailov et al., 2023).

## 3 Data Collection

In this paper, we focus on data from Wikipedia and Wikinews, both under the Wikimedia Foundation.

Wikipedia uses a hierarchical classification system for articles. It begins with top-level categories that cover broad fields, which are then divided into more specific subcategories. Only pages four or five levels away from our target category are included in our study. Then we scrape the Wikipedia page versions from 2018 to 2025, using the January 1 snapshot of each year. To minimize the impact of topic-specific words, only those ranked within the top 10,000 in the Google Ngram dataset[2] are included in the calculations.

We are interested in Wikipedia pages that belong to the following categories: *Art*, *Biology*, *Computer Science (CS)*, *Chemistry*, *Mathematics*, *Philosophy*, *Physics*, *Sports*. Among them, *Philosophy* has the smallest number of articles (31,132), and *CS* leads with the largest number (55,121). More details on data collection and processing are shown in Appendix A. For a better comparison, we also collect 6,690 *Featured Articles (FA)*, along with their corresponding 2,029 simple English versions (where available) as *Simple Articles (SA)*.

While Wikipedia is the main focus of this paper, we also collect Wikinews articles from 2020 to 2024 to generate questions in Section 5.2. There are over a hundred news per year, covering a wide variety of topics.

## 4 Direct Impact from LLMs

### 4.1 Direct Impact 1: Word Frequency

Since LLMs are extensively applied to writing-related tasks, we aim to investigate whether the text in Wikipedia articles has changed as well. For example, we found that the frequency of certain words favored by LLMs has indeed increased, such as "*crucial*" and "*additionally*" (Geng & Trotta, 2024; Kobak et al., 2024), as shown in Figure 2.

To further investigate whether the changes in word frequency are coincidental or part of a collective shift, we calculate the frequency changes of more words and estimate the impact of LLMs $\eta$ in one set of Wikipedia articles $S$ based on the following formula (Geng et al., 2025):

$$\hat{\eta}(S) = \frac{\sum_{i \in I} \left( f_i^d(S) - f_i^*(S) \right) f_i^*(S) \hat{r}_i}{\sum_{i \in I} \left( f_i^*(S) \hat{r}_i \right)^2}, \tag{1}$$

$$\hat{r}_i = \frac{f_i(S_2) - f_i(S_1)}{f_i(S_1)}, \tag{2}$$

where $f_i^d(S)$ represents the frequency of word $i$ in the set of texts $S$ in real dataset, $f_i^*(S)$ represents the one if LLMs do not affect the texts, $I$ is the set of words used for estimation, $f_i(S_1)$ and $f_i(S_2)$ represent the frequency of word $i$ for another set of articles before and after LLM processing, respectively.

---

[2]Google Ngram dataset: `https://www.kaggle.com/datasets/wheelercode/english-word-frequency-list`

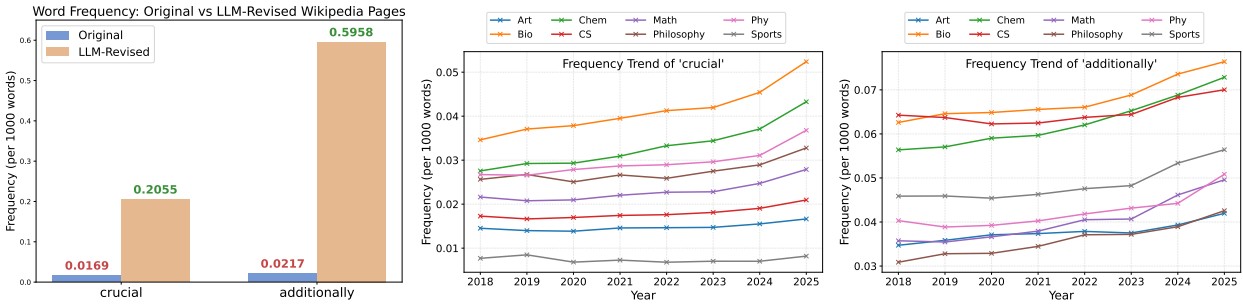

Figure 2: Word frequency before and after LLM processing, and its evolution in Wikipedia articles.

We take the average of the word frequencies from the 2018, 2019, and 2020 versions of the pages as $f_i^*(S)$. To construct word frequency data $f(S_2)$ reflecting the impact of LLMs, we use *GPT-4o-mini* to revise the January 1, 2022, versions of *Featured Articles*, with the prompt *"Revise the following sentences:"*.

By setting thresholds for $f^*$ and $\hat{r}$, we can select commonly-used and LLM-sensitive word combinations $I$ to estimate the impact of LLMs. We perform a grid search over the parameter space used for vocabulary selection. Specifically, $f^*$ ranges from 500 to 20,000 with a step size of 500, and $\hat{r}$ ranges from 0.05 to 1.0 with a step size of 0.05. For each $(f^*, \hat{r})$ combination, we begin by estimating the LLM impact over the pre-LLM period (2018–2022). Using only these pre-LLM estimates, we fit a linear regression to model the natural evolution trend of Wikipedia articles.

We then evaluate each parameter combination based on two criteria: (i) how well the linear model fits the data, measured by the coefficient of determination $R^2$, and (ii) the stability of the pre-LLM baseline, measured by the absolute value of the fitted slope. Parameter combinations with $R^2$ close to 1 and slopes close to 0 indicate a good baseline before LLM adoption. To avoid reliance on a single criterion, we select the intersection of the $TOP_K$ parameter combinations ranked by $R^2$ and by slope.

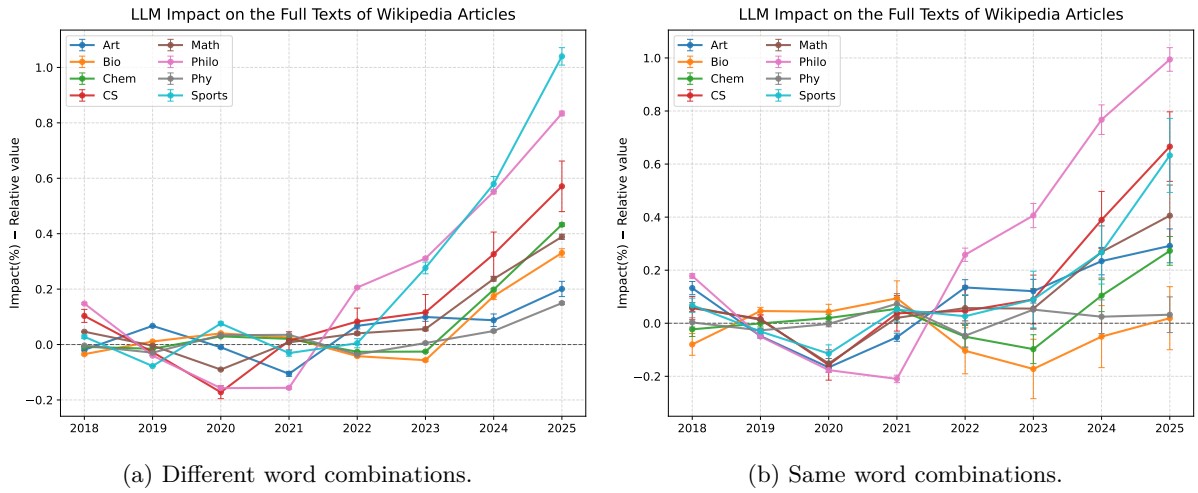

(a) Different word combinations.

(b) Same word combinations.

Figure 3: Impact of LLMs on Wikipedia pages, estimated based on simulations of *Featured Articles*.

For the $f^*$ value, we propose two strategies: First, the target words should frequently appear in the first section of *Featured Articles*, as we use this part of the articles for LLM simulation when estimating $\hat{r}$; second, the target words should frequently appear in the target category. For the first strategy, when calculating the impact of the LLM on different pages, the selected vocabulary combination remains the same. For the second strategy, the influence on pages of different categories will be estimated using the vocabulary combination corresponding to each category.

Finally, we use the selected words to estimate impacts for the post-LLM period (2023–2025). We subtract the extrapolated pre-LLM linear trend from the estimates to separate causality. When setting $TOP_K = 250$, the LLM impact is approximately 1% for the articles in certain categories, as illustrated in Figures 3. But different texts still lead to different estimations, and using different words for estimation will also produce different results. Detailed results of parameter selection and impact estimation are shown in Appendix B.

> **Finding 1:** While the estimation results vary, the influence of LLMs on Wikipedia is likely to become more significant over time. In some categories, the impact has exceeded 1%.

## 4.2 Direct Impact 2: Linguistic Style

We also investigate the current and future impact of LLMs on Wikipedia from more linguistic perspectives, examining the evolution of Wikipedia content at **Word**, **Sentence**, and **Paragraph** levels by comparing the texts before and after LLM processing under the same standards.

| Criteria | Effect of LLM processing | Trends in real data | Figures |
|---|:---:|:---:|---|
| Auxiliary Verb Rate % | ↘ | ↘ | 4a, 4d |
| To be Verb Rate % | ↘ | ↘ | 15 |
| Corrected Type-Token Ratio (CTTR) | ↗ | ↗ | 16 |
| Long Words Rate % | ↗ | ↗ | 17 |
| Conjunction Rate % | – | ↗ | 18 |
| Noun Rate % | ↗ | ↗ | 19 |
| Preposition Rate % | – | ↗ | 20 |
| Pronouns Rate % | ↘ | – | 21 |
| One-syllable Word Rate % | ↘ | ↘ | 22 |
| Average Syllables per Word | ↗ | ↗ | 23 |
| Passive Voice Rate % | ↘ | ↗ | 4b, 4e |
| Long Sentence Rate % | ↗ | ↗ | 24 |
| Average Sentence Length | ↗ | ↗ | 25 |
| Average Parse Tree Depth | ↗ | ↗ | 26 |
| Clause Rate % | ↗ | ↗ | 27 |
| Pronoun-initial Sentence Rate % | ↘ | ↗ | 28 |
| Article-initial Sentence Rate % | – | ↗ | 29 |
| Dale-Chall Readability | ↗ | ↘ | 4c, 30a |
| Automated Readability Index | ↗ | ↗ | 4c, 30b |
| Flesch-Kincaid Grade Level | ↗ | ↗ | 4c, 4f |
| Flesch Reading Ease | ↘ | – | 4c, 30c |
| Coleman-Liau Index | ↗ | – | 4c, 30d |
| Gunning Fox Index | ↗ | ↗ | 4c, 30e |

Table 1: Summary of linguistic style trends. The second column indicates the effects of LLM processing. The third column shows Wikipedia trends over time.

**Word Level.** In addition to word frequency used before, we can also consider it from a broader perspective at the level of words. For instance, the *frequency of auxiliary verbs* indicates the ability of a model to convey complex reasoning and logical relationships (Yang et al., 2024). Lexical diversity, often measured by the *corrected type-token ratio (CTTR)*, reflects the variety of words (Wróblewska et al., 2025). Herbold et al. (2023) revealed that the lexical diversity of humans is higher than that of ChatGPT-3 but lower than that of ChatGPT-4, suggesting newer models have surpassed human writing in that metric. Furthermore, the

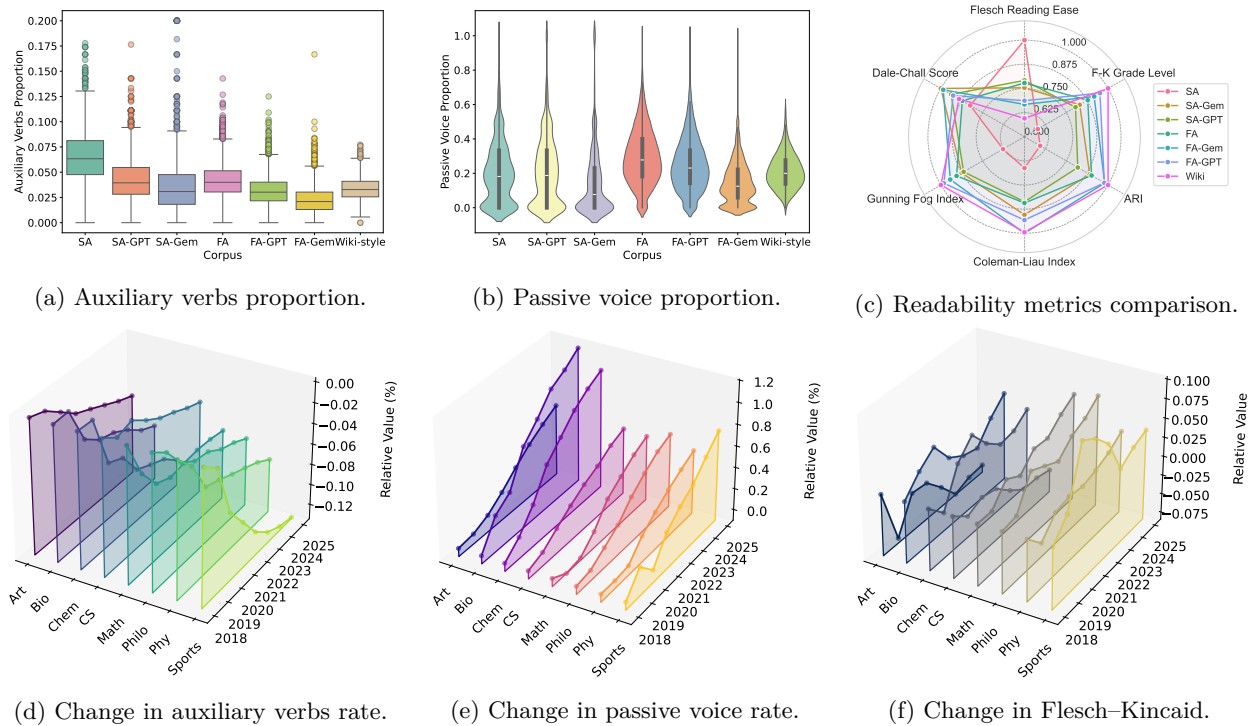

(a) Auxiliary verbs proportion.

(b) Passive voice proportion.

(c) Readability metrics comparison.

(d) Change in auxiliary verbs rate.

(e) Change in passive voice rate.

(f) Change in Flesch–Kincaid.

Figure 4: The results of linguistic style comparison, including the real Wikipedia pages and LLM-simulated pages. The three subplots below represent the differences compared to the data from 2020.

*proportion of specific parts of speech (POS)* is commonly used as a stylistic feature in assessing the quality of Wikipedia articles (Moás & Lopes, 2023). Georgiou (2025) showed that LLM-generated text employed more nouns, while human-written text employed more auxiliaries and pronouns.

**Sentence Level.** In terms of sentence structure, we focus on *sentence length* and the use of *passive voice* (AlAfnan & MohdZuki, 2023). Regarding sentence complexity, we analyze both the *depth of the entire syntactic tree* and the *clause ratio* (Iavarone et al., 2021). Reinhart et al. (2025) revealed that LLMs use present participial clauses at 2 to 5 times the rate of human text while using the passive voice at roughly half the rate as human texts.

**Paragraph Level.** For the paragraph dimension, which is essential for Wikipedia's educational mission (Johnson et al., 2024b), we seek guidance from *readability* evaluation (Moás & Lopes, 2023; Trokhymovych et al., 2024), where six traditional formulas are included in our study: *Automated Readability Index* (Mehta et al., 2018), *Coleman-Liau Index* (Antunes & Lopes, 2019), *Dale-Chall Score* (Patel et al., 2011), *Flesch Reading Ease* (Eleyan et al., 2020), *Flesch–Kincaid Grade Level* (Solnyshkina et al., 2017), and *Gunning Fog index* (Świeczkowski & Kułacz, 2021). The detailed definitions of all metrics are provided in Appendix C.3.

**LLM Simulation.** Wikipedia articles are not static, and their linguistic styles are difficult to remain the same under different measurement metrics. To understand the link between these trends and LLMs, we simulate the real Wikipedia with *GPT-4o-mini* and *Gemini-1.5-Flash*, then compare the changes before and after the process. Specifically, we instruct both models to revise *Featured* and *Simple* articles using prompt mentioned in Section 4.1, and additionally use *GPT-4o-mini* to generate Wikipedia-style articles using prompt *"Generate a Wikipedia-style article titled {title} and return only the article body in plain text."*

**Results.** Table 1 presents the summary of the trends in linguistic style in real Wikipedia articles and LLM simulations. The detailed outcomes are illustrated in Figure 4 and Appendix C. Although we have plotted the results from 2020 in the these figures, the trends summarized in the table are based on the data in the LLM era, that is, after 2023. For example, our simulation results reveal that LLMs substantially reduce the use of *auxiliary verbs*, with Gemini employing even fewer than GPT, as shown in Figure 4a. Consistent with this tendency, the usage of *auxiliary verbs* on real Wikipedia pages shows a marginal decline from 2020 to 2025, as depicted in Figure 4d. However, this consistency is not observed for all metrics. For instance, the trends of *passive voice proportion* in Figures 4b and 4e are not the same. For *paragraph level*, Figure 4c presents the results of six *readability* metrics, all of which indicate that LLM-generated texts tend to be less readable. The Flesch–Kincaid score in Figure 4f initially decreases and then rises, and the score after LLM simulation also increases.

> ***Finding 2:*** The observed changes in Wikipedia articles are largely consistent with the preferences of LLMs under most metrics.

### 4.3 Direct Impact 3: Page View

The analysis of Wikipedia's page view data (*i.e.*, the number of times a Wikipedia page is accessed by users) can yield many interesting conclusions (Piccardi et al., 2021; 2024). Similar to the work of Reeves et al. (2024), we transform the page view counts of Wikipedia articles using the inverse hyperbolic sine function.

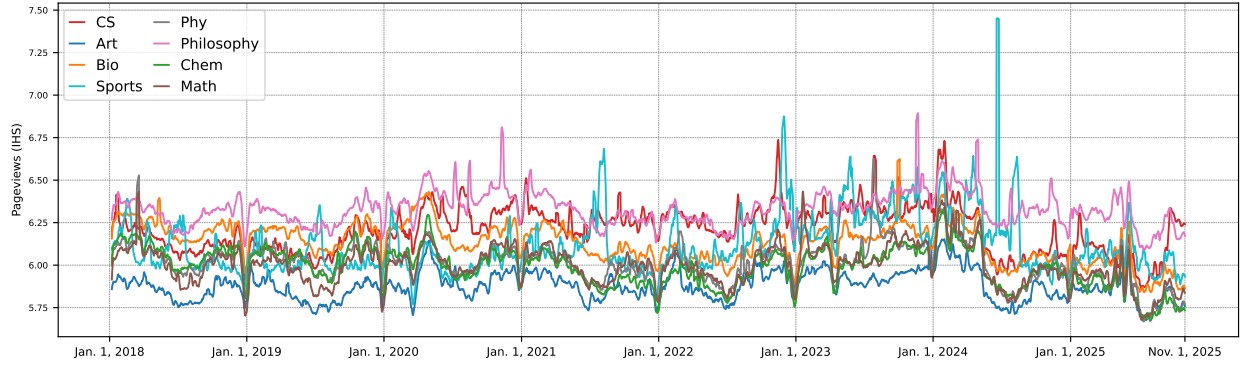

(a) Page views Across Different Categories in English Wikipedia.

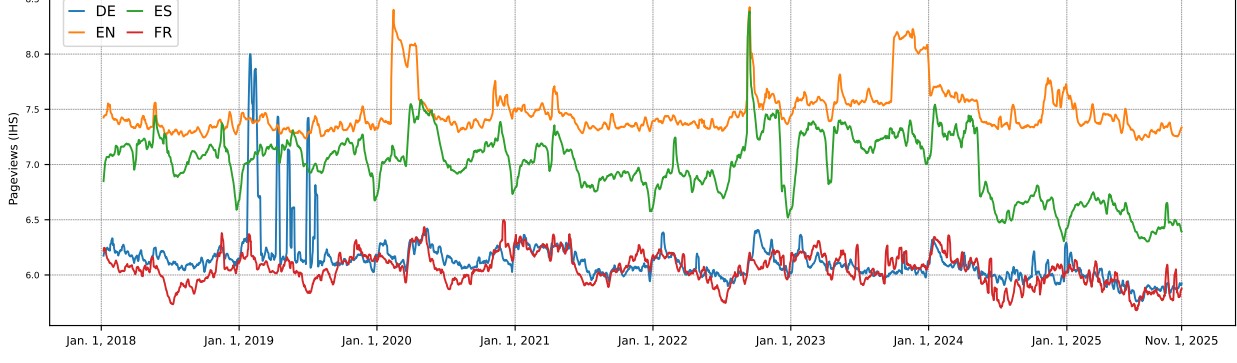

(b) Page views Across Wikipedia of Different Languages

Figure 5: Daily page views of Wikipedia pages. The y-axis represents page view values after smoothing with a seven-day time window and being transformed via the *Inverse Hyperbolic Sine (IHS)* function.

Figure 5a shows page views across different categories in English Wikipedia. Notably, there was a slight decline in page views across some scientific categories since 2024. Reeves et al. (2024) examined changes in

Wikipedia page views across various languages from up to January 1, 2024, but our analysis covers pages from different scientific categories and extends to 2025. The latest data actually leads to different findings, and one recent study has reached a similar conclusion (Lyu et al., 2025).

To generalize our findings beyond English (EN) Wikipedia, we further analyze the page views of *Featured Articles* in four major language editions, German (DE), Spanish (ES) and French (FR), as shown in Figure 5b. The page views of Wikipedia articles in these languages also exhibit a decline from 2024, with the drop being especially sharp in the Spanish edition. Detailed data for different language editions is shown in Table 3, and the means of the page view values are plotted in Figures 31 in the appendix.

> **Finding 3:** In the second half of 2024, there was a slight decline in page views across some scientific categories, and its connection to the use of LLMs requires further investigation.

# 5 Indirect Impact from LLMs

## 5.1 Indirect Impact 1: Machine Translation

**Overall.** Sentences of some machine translation benchmarks are derived from Wikipedia. If these benchmarks are also influenced by LLMs, what impact would it have on the evaluation results?

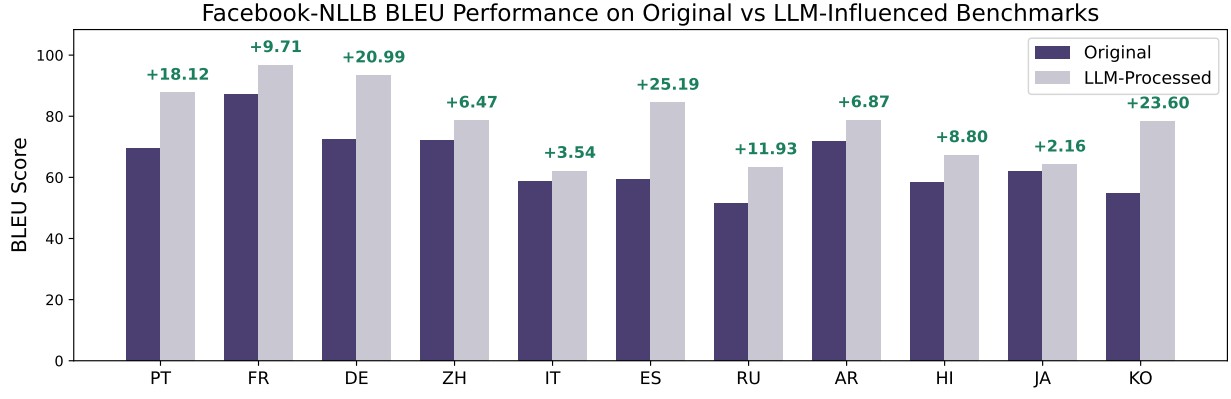

Figure 6: Facebook-NLLB BLEU scores on the original benchmark and the LLM-Influenced benchmark.

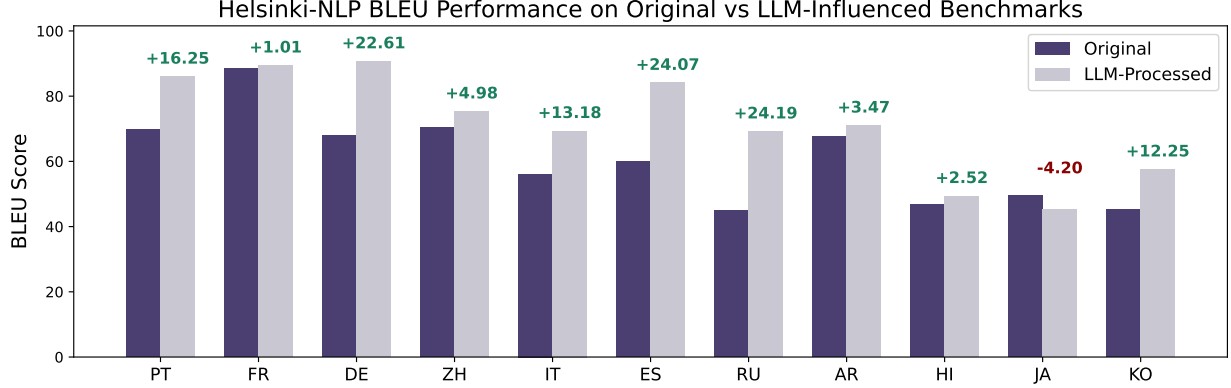

Figure 7: Helsinki-NLP BLEU scores on the original benchmark and the LLM-Influenced benchmark.

**Benchmark Construction.** We utilize the *Flores dataset*[3], which consists of multiple sentence sets, each containing parallel translations of the same Wikipedia sentence across many languages. For our experiments, we keep the original English (*EN*) sentence in each set, and use *GPT-4o-mini* to translate this English sentence into the remaining languages with the prompt *"Translate the following text to {target language}"*. We then replace the original non-English sentences with these LLM-generated translations, forming an LLM-influenced version of the benchmark. The following 11 widely used languages are included in our simulations: Modern Standard Arabic (*AR*), Mandarin (*ZH*), German (*DE*), French (*FR*), Hindi (*HI*), Italian (*IT*), Japanese (*JA*), Korean (*KO*), Brazilian Portuguese (*PR*), Russian (*RU*), Latin American Spanish (*ES*).

**Evaluation Pipeline.** We evaluate multiple machine translation models by translating the benchmark sentences and assessing the output with four metrics: BLEU (Post, 2018), COMET (Rei et al., 2020), ChrF (Popović, 2015), and BERTScore (Zhang et al., 2019).

**Models.** We compare the translation results from three models: *Facebook-NLLB*[4], a multilingual model supporting 200+ languages (Costa-Jussà et al., 2022); *Google-T5 (mT5)*[5], pre-trained on data covering 101 languages (Xue et al., 2021); and *Helsinki-NLP*'s bilingual Transformer models[6] trained on OPUS corpus (Tiedemann & Thottingal, 2020; Tiedemann et al., 2023).

**Results.** In most cases, machine translation models achieve higher scores on the GPT-processed benchmark, as shown in Figures 6 and 7. Moreover, the relative ranking of machine translation models could be reversed. For example, in the French translation task, *Facebook-NLLB* gets a lower BLEU score (87.04) than *Helsinki-NLP* (88.39) in the original benchmark, but a better score in the GPT-processed benchmark (96.75 *vs* 89.40). More results are shown in Figure 32, 33, 34, and 35 in the Appendix.

> *Finding 4:* The impact of LLMs on the benchmark could not only inflate the translation scores across different languages but also distort the comparison of translation abilities between models, thereby undermining the benchmarks' ability to reflect true translation effectiveness.

## 5.2 Indirect Impact 2: RAG

**Overall.** RAG can provide more reliable and up-to-date external knowledge to mitigate hallucination in LLM generation (Gao et al., 2023). Wikipedia is one of the most commonly applied general retrieval sets in previous RAG work, which stores factual structured information in scale (Fan et al., 2024). In the process of translation using LLMs, some information may also be lost or distorted (Mohamed et al., 2025). Therefore, we are curious how the effectiveness of RAG might change if Wikipedia pages are influenced by LLMs. Our experiment procedure is illustrated in Figure 8 and the detailed steps are listed below.

**Question Generation.** *GPT-4o-mini* and *Gemini-1.5-flash* are used to generate multiple-choice questions (MCQs) based on Wikinews articles. In order to generate some questions that are not too easy for LLMs, we refer to the prompt in the work of Zhang et al. (2025b), shown in Figure 9.

**Knowledge Base.** We construct the knowledge base using Wikinews articles from 2020 to 2024. Each article is preprocessed and split into smaller text segments, then vectorized via BERT[7] (Devlin et al., 2019). We then indexed these vectors using FAISS, a library for efficient similarity search and clustering of dense vectors, to enable rapid retrieval (Douze et al., 2024).

**Retrieval and Generation.** The questions are vectorized using BERT and a similarity search is conducted with FAISS. The three most relevant segments are retrieved and used as contextual information. These

---

[3]https://huggingface.co/datasets/openlanguagedata/flores_plus
[4]https://huggingface.co/facebook/nllb-200-3.3B
[5]https://huggingface.co/google/mt5-small
[6]https://github.com/Helsinki-NLP/Opus-MT
[7]https://huggingface.co/bert-base-uncased

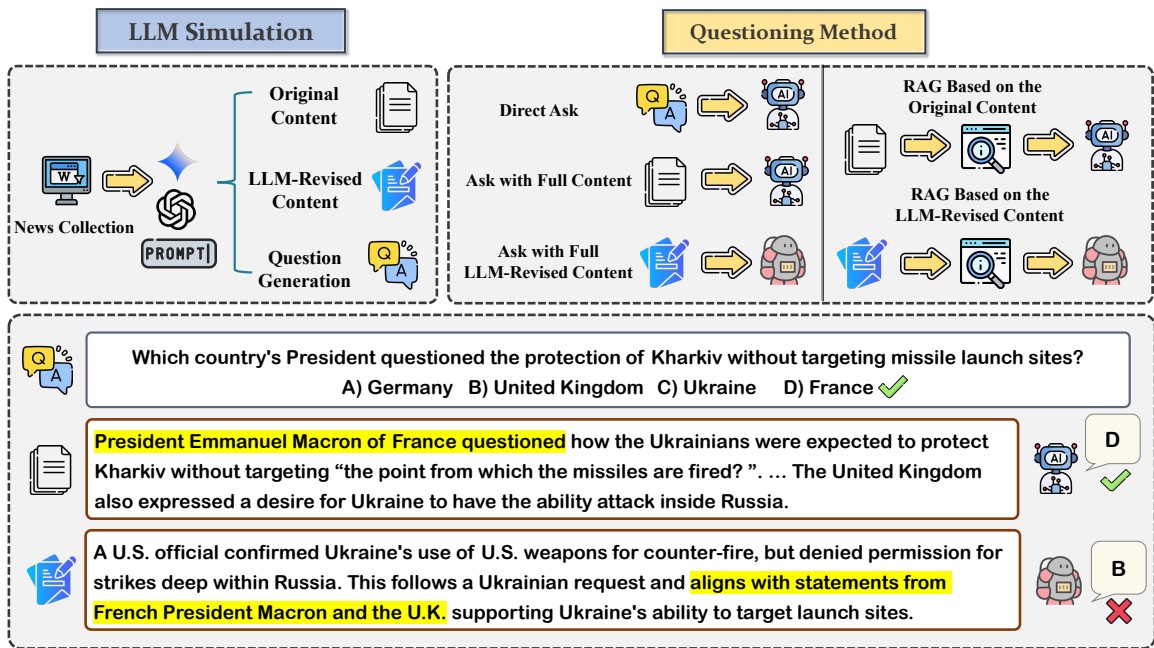

Figure 8: *GPT-4o-mini* and *Gemini-1.5-flash* are used to generate multiple-choice questions (MCQs) based on the extracted Wikinews data. Various questioning methods are employed to evaluate the impact of LLMs on the RAG process. The LLMs being queried include *GPT-4o-mini*, *GPT-3.5*, and *DeepSeek-V3*.

---

**Prompt**

You are to generate three self-contained multiple-choice questions based on the facts mentioned in the following content. Avoid questions that reference the content directly. Each question should include all relevant context and directly name any referenced items, avoiding pronouns like *"it," "the game,"* or *"the person."* Do not include phrases that reference the source or context, such as *"mentioned in the article"* or *"according to the text."*

---

Figure 9: Prompt for Wikinews-based questions.

segments are then combined with the question in a prompt template to query *GPT-3.5* and *GPT-4o-mini*. The final answer is generated based on both the LLMs' prior knowledge and the retrieved content.

**Questioning Methods.** We conduct experiments using three types of queries. First, we can query the LLMs directly to obtain answers, using the prompt shown in Figure 36 (**Direct Ask**). Second, the Wikinews articles used to generate the question is included in the prompt, as shown in Figure 37. To explore the impact of LLM-generated text, we apply the prompt *"Revise the following news"* to Wikinews articles. We consider three versions of each article: the original text (**Full (Original)**), and revisions processed by GPT-4o-mini (**Full (GPT)**) or Gemini-1.5-Flash (**Full (Gemini)**). Finally, in the RAG-based setting, relevant information is first retrieved from a constructed knowledge base, after which the models are queried using the prompt shown in Figure 10. The knowledge base can be built from original Wikinews articles (**RAG (Original)**) or from articles revised by GPT-4o-mini (**RAG (GPT)**) or Gemini-1.5-Flash (**RAG (Gemini)**).

**Results.** Figure 11 illustrates the summary of the accuracy rates of the LLM responses under different scenarios, with more detailed results provided in Appendix F. The analysis based on these results leads to the following conclusions:

---

**Prompt: Asking with a knowledge base**

```
prompt = (
    f"Use context to answer user questions."
    f"Question:  {question}\n"
    f"Reference context:  {topk_ans}\n"
    f"Only need to give the correct option without explanation."
)
```

---

Figure 10: Prompt used in the asking with a knowledge base setting.

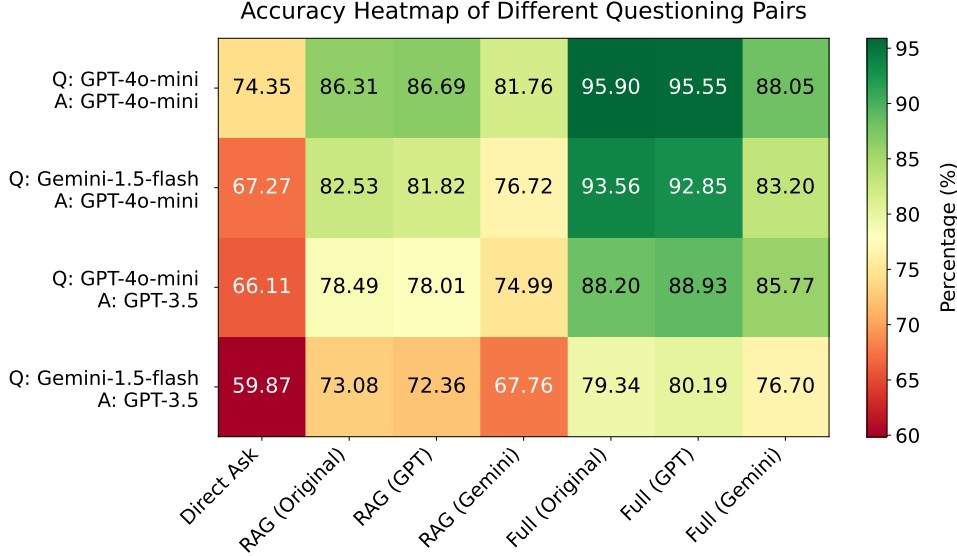

Figure 11: The accuracy rate of LLM responses under different settings. For each case, more than 1,800 questions based on Wikinews articles from 2020 to 2024 are used for simulations. More detailed results are presented in Appendix F.

- **Higher Accuracy with Knowledge Base.** Providing external knowledge greatly improves performance. With a knowledge base, the accuracy of responses often exceeds 80%. This confirms the effectiveness of RAG in enhancing factual accuracy.

- **Maximal Performance with Full Content.** Providing the full news as context yields the highest accuracy, demonstrating the limitations of retrieval-based approaches in selecting the most relevant information. In most cases with *GPT-4o-mini*, the full content approach achieves more than 93% accuracy, setting a benchmark for ideal retrieval performance.

- **Impact of LLM-Revised Content.** Compared to the cases using real Wikinews articles, the accuracy of responses based on ChatGPT-processed pages shows little change and responses based on Gemini-processed pages show a clear drop in accuracy. This suggests that Gemini's rewriting may lead to the loss of some key information.

- **Declining Accuracy for Recent Events.** In the absence of RAG, all models exhibit lower accuracy when answering questions derived from recent Wikinews articles (*e.g.*, *GPT-4o-mini* shown in Table 10 of the appendix: 66.67% in 2024, *GPT-3.5*: 61.25% in 2024), while their accuracy is much better for older events (*e.g.*, 2020–2022). The reason is also straightforward: these news events are not included in their training data. Moreover, DeepSeek-V3 achieves the highest accuracy, which may be attributed to its later knowledge cutoff date. In addition, Table 10 also report results rewritten using the newly released *Gemini-3*. Compared to outputs rewritten with *Gemini 1.5*, both **RAG (Gem3)** and **Full**

**(Gem3)** exhibit performance improvements. This suggests that as LLMs continue to advance, the risk of information loss introduced by LLM-based rewriting may be partially mitigated.

**Case Study.** To explore the impact of LLM-generated texts, we focus on cases in which the model answers correctly with the original content but fails with the LLM-revised version. Figure 8 has provided one interesting example. The original passage[8] contained an unambiguous clue, "President Emmanuel Macron of France questioned how the Ukrainians were expected to protect Kharkiv ...," which directly supports the correct answer "France." However, in the LLM-revised version, the model reformulated the information into a more abstract and compressed form: "...aligns with statements from French President Macron and the U.K. ..." This revision removes the explicit verb "questioned," merges multiple entities, and relocates key details. As a result, RAG systems relying on the revised text may incorrectly associate the query with "the U.K." due to lexical proximity. This illustrates how LLM-style rewriting can distort relational information and impair factual grounding in RAG systems. More examples are included in Appendix F.3, and LLM-generated texts may decrease accuracy in RAG tasks for several reasons:

- **Information Fusion Misleading**: When LLMs merge multiple distinct and clear pieces of information into a single sentence, it can lead to misinterpretation as shown in Figure 8.
- **Keyword Replacement and Omission**: LLM might replace or omit key terms, altering the original meaning and causing misinterpretation in Figures 38, 39 and 40.
- **Abbreviation Ambiguity Misleading**: LLMs use abbreviations inappropriately, leading to misinterpretation as shown in Figure 41.
- **Introduction of Modifiers**: Adding adjectives or modifiers can change the context and impact the text's accuracy, as illustrated in Figure 42.
- **Retrieval Mismatch**: Revised texts may either reduce the similarity between the question and the correct news or increase the similarity with irrelevant ones. In some cases, even small edits to the article lead to a failure in matching.

> ***Finding 5:*** The results suggest that LLM-processed content may perform less effectively in RAG systems compared to human-created texts in some cases. If such content has impacted high-quality communities like Wikipedia, it raises concerns about the potential decline in information quality in knowledge bases.

## 6 Discussion and Conclusion

The relationship between Wikipedia and LLMs is bidirectional. On the one hand, Wikipedia content has been a key factor in the growth of LLMs. On the other hand, researchers have used NLP methods, including LLMs, to improve Wikipedia (Lucie-Aimée et al., 2024). Humans and LLMs are coevolving (Geng & Trotta, 2025), and Wikipedia may be one of the bridges in this process. Our study also provides new insights into the risks associated with research that uses Wikipedia data.

In this paper, we collect a large amount of real-world data and conduct comprehensive experimental simulations. Our findings suggest that LLMs are impacting Wikipedia and the impact could extend indirectly to some NLP tasks through their dependence on Wikipedia content. For instance, the target language for machine translation may gradually shift towards the language style of LLMs, albeit in small steps. In addition, the accuracy of RAG tasks may decline when LLM-revised Wikipedia pages are used, indicating potential risks of using LLMs to support Wikipedia or similar knowledge systems.

Although some of the changes may not be immediately apparent, our work offers a framework for long-term monitoring. These results will also serve as excellent illustrations of the impact of AI on society, given the significant amount of human engagement with Wikipedia. This kind of social impact is already taking place, but it may not have been adequately addressed by the AI community.

---

[8]https://en.wikinews.org/wiki/Ukraine_permitted_to_strike_Russian_territory_near_Kharkiv

## Limitations

Although we conduct several experiments to evaluate the impact of LLMs on Wikipedia, our study has certain limitations. First, some analyses are primarily correlational, identifying patterns but not definitively attributing observed changes to LLMs. The causal relationships of some impacts, such as the pages views, require more detailed discussion.

Second, the lack of field experiments limits our insights into the actual human-in-the-loop editing processes behind Wikipedia article creation. Real-world editing involves complex interactions between humans and sophisticated LLM-based tools. These dynamics may not be fully captured by our simulated studies.

Additionally, when assessing the readability of Wikipedia pages, we rely only on traditional metrics based on formulas, such as the Flesch-Kincaid score. However, recent advances in NLP have shifted towards computational models (François, 2015). Moreover, in the RAG task, our Wikinews dataset is not large enough compared to the Wikipedia page dataset, which may limit the generalization of our findings.

## Acknowledgments

This work benefited from funding from the French State, managed by the Agence Nationale de la Recherche, under the France 2030 program (grant reference ANR-23-IACL-0008). This work is also supported in part by the ENS-PSL BeYs Chair in Data Science and Cybersecurity.

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

## A  Data Collection and Processing

The detailed classification in Wikipedia poses a problem in our data crawling process: When iteratively querying deeper subcategories without limit, the retrieved pages may become less relevant to the original topic (*i.e.*, the root category). To address this issue, we select an appropriate crawl depth for each category to balance the number of pages with their topical relevance, as shown in Table 2.

We also exclude redirect pages, as they do not contain independent content but link to other target pages. After crawling the pages, we clean the data by extracting the plain text and removing irrelevant sections such as *"References," "See also," "Further reading," "External links," "Notes,"* and *"Footnotes."* For Wikinews, we use the *TextExtracts extension*[9], which provides an API to retrieve plain-text extracts of page content.

---

[9]TextExtracts extension: `https://www.mediawiki.org/wiki/Extension:TextExtracts#query+extracts`

| Category | Art | Bio | Chem | CS | Math | Philo | Phy | Sports |
|---|---|---|---|---|---|---|---|---|
| **Crawl Depth** | 4 | 4 | 5 | 5 | 5 | 5 | 5 | 4 |
| **Number of Pages** | 50,810 | 41,237 | 49,516 | 55,121 | 43,888 | 31,132 | 38,144 | 48706 |

Table 2: Number of Wikipedia articles crawled per category.

| Languages | German | Spanish | English | French |
|---|---|---|---|---|
| **Number of Pages** | 2943 | 1363 | 6690 | 2254 |

Table 3: Number of Wikipedia Featured Articles in different language.

## B  LLM Impact

Let $\tau_f$ denote the threshold for the baseline frequency $f_i^*$, and let $\tau_r$ denote the threshold for the sensitivity measure $\hat{r}_i$. We define a discrete search space $\Theta$ as follows:

$$\Theta = \{(\tau_f, \tau_r) \mid \tau_f \in \mathcal{T}_f, \tau_r \in \mathcal{T}_r\} \tag{3}$$

where $\mathcal{T}_f = \{500, 1000, \ldots, 20{,}000\}$ represents the sequence of frequency thresholds, and $\mathcal{T}_r = \{0.05, 0.10, \ldots, 1.0\}$ represents the sequence of sensitivity thresholds.

For each parameter pair $\theta = (\tau_f, \tau_r) \in \Theta$, we construct a candidate vocabulary set $I_\theta$. This set consists of words that are both frequently used in the target texts and sensitive to LLMs, i.e., words that are either particularly preferred or clearly avoided by LLMs:

$$I_\theta = \left\{ w \in \mathcal{V} \ \middle| \ \frac{1}{f_w^*} \leq \tau_f \ \wedge \ \frac{\hat{r}_w + 1}{\hat{r}_w^2} \geq \frac{\tau_r + 1}{\tau_r^2} \right\} \tag{4}$$

To evaluate the reliability of the candidate set $I_\theta$, we examine the natural evolution trend of the indicator $\hat{\eta}$ during the pre-LLM period, defined as $T_{pre} = \{2018, \ldots, 2022\}$. We model this trend using linear regression, where the slope of the fitted line is denoted by $\alpha_\theta$.

To avoid bias from reliance on a single metric, let $\mathcal{S}_{R^2}$ and $\mathcal{S}_\alpha$ denote the $\text{TOP}_K$ parameter sets ranked by descending $R^2$ and ascending $|\alpha_\theta|$, respectively. The optimal parameter set $\Theta^*$ is defined as their intersection:

$$\Theta^* = \mathcal{S}_{R^2} \cap \mathcal{S}_\alpha \tag{5}$$

When setting $TOP_K = 250$ to estimate the full texts of Wikipedia articles, 18 $(f^*, \hat{r})$ combinations that satisfy the conditions are: (5500, 0.45), (6000, 0.45), (5000, 0.4), (4500, 0.35), (5000, 0.35), (7500, 0.45), (7000, 0.45), (6500, 0.45), (8000, 0.45), (8500, 0.5), (9000, 0.5), (9500, 0.5), (15000, 0.5), (10000, 0.5), (15500, 0.5), (10500, 0.5), (16500, 0.5), (17500, 0.5).

Specifically, when we take $\frac{1}{f^*} \leq 5500$ and $\frac{\hat{r}+1}{\hat{r}^2} \geq \frac{0.45+1}{0.45^2}$, 115 words that satisfy the conditions are: "moved," "run," "called," "players," "taken," "largely," "seen," "struck," "remains," "mainly," "press," "make," "appeared," "long," "launched," "sometimes," "earlier," "like," "form," "wide," "player," "sent," "subsequently," "brought," "had," "upon," "despite," "significant," "killed," "making," "us," "can," "given," "parts," "leading," "see," "came," "primarily," "important," "throughout," "worked," "failed," "this," "p," "very," "saw," "large," "due," "features," "usually," "just," "however," "attempt," "built," "different," "because," "victory," "popular," "men," "across," "commonly," "out," "there," "placed," "mostly," "went," "particularly," "serving," "often," "having," "following," "operations," "died," "established," "wrote," "forced," "so," "almost," "where," "but," "whose," "lived," "next," "helped," "served," "various," "generally," "soon," "while," "number," "written," "win," "people," "initially," "considered," "used," "these," "rest," "along," "located," "won," "role," "limited," "numerous," "use," "fought," "about," "result," "opened," "up," "subsequent," "then," "ended," "caused," "within."

Figures 12, 13, and 14 illustrate LLM impact estimated using different $TOP_K$.

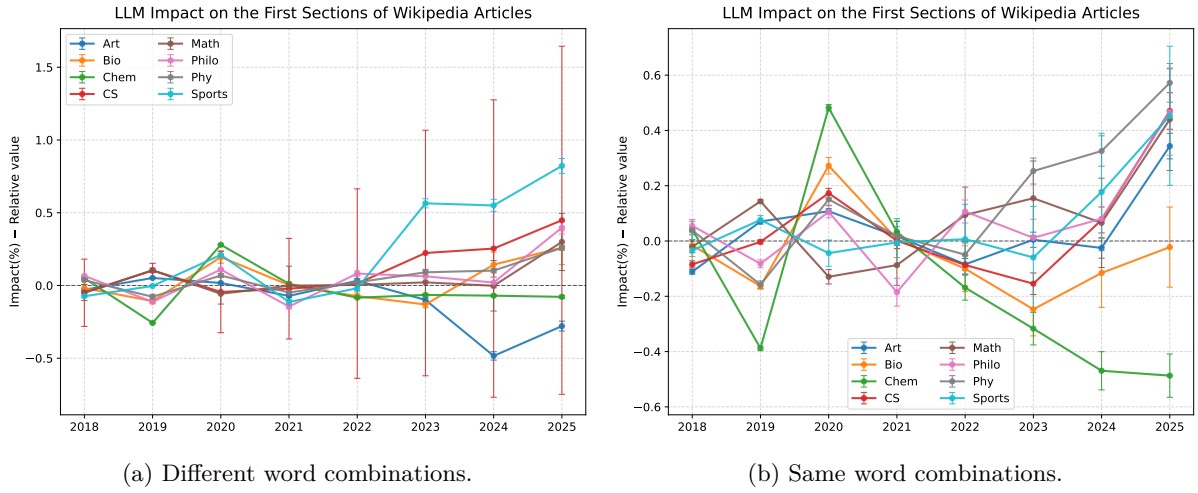

Figure 12: Impact of LLMs on the first section of Wikipedia pages, estimated when setting $TOP_K = 250$.

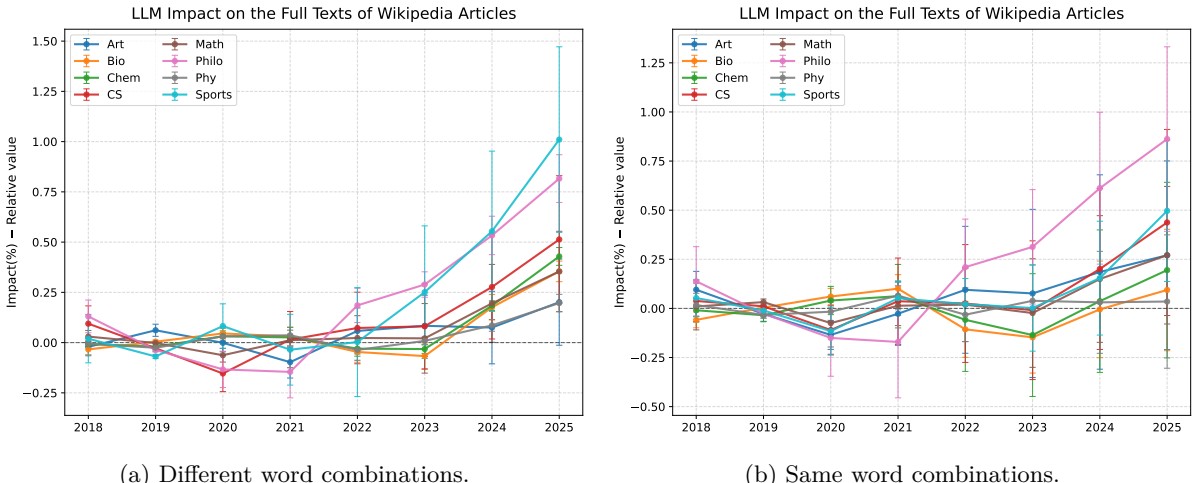

Figure 13: Impact of LLMs on the full texts of Wikipedia pages, estimated when setting $TOP_K = 300$.

## C  Linguistic Style

### C.1  Word Level

**"To Be" Verbs**  Figure 15 illustrates that LLMs significantly reduce the usage of *"To Be"* verbs (*e.g.*, replacing *"is important"* with *"demonstrates significance"*), with Gemini using fewer such verbs than GPT. Moreover, a marginal decline in the usage of these verbs is observed in actual Wikipedia pages.

**Lexical Diversity**  As shown in Figure 16, revised articles display a slightly higher *CTTR*, with texts revised by GPT exhibiting greater lexical diversity than those revised by Gemini. When tasked with generating wiki-style articles, GPT achieves the highest lexical diversity. Over time, the vocabulary used across different Wikipedia categories has become increasingly varied.

**Long Words**  Figure 17 shows that LLM-revised texts generally contain a higher proportion of long words than human-written pages, with Gemini producing the most pronounced increase. From 2020 to 2025, a substantial increase in the usage of long words is observed in the first section of Wikipedia pages.

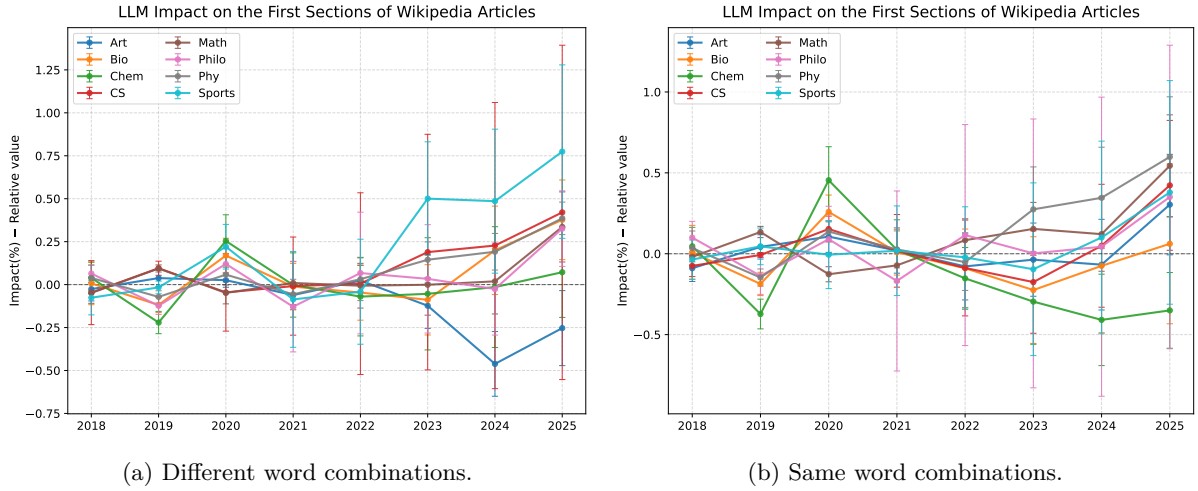

(a) Different word combinations.

(b) Same word combinations.

Figure 14: Impact of LLMs on the first section of Wikipedia pages, estimated when setting $TOP_K = 300$.

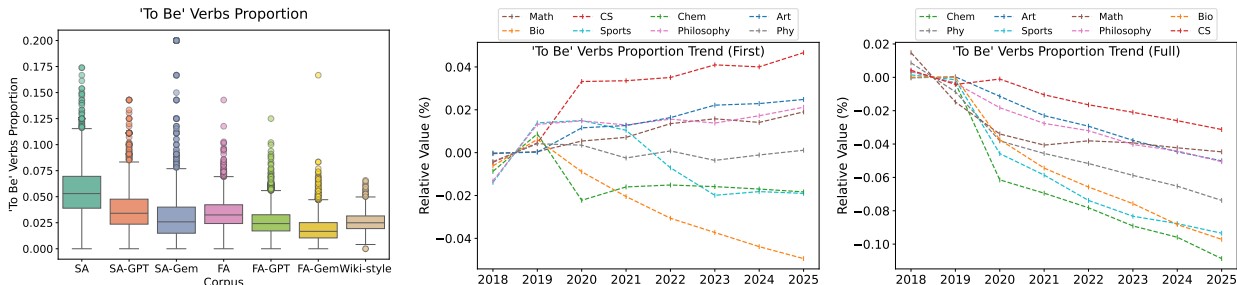

Figure 15: *"To Be"* Verbs.

**Parts of Speech**  Figures 18, 19, 20, and 21 show that LLMs lead to a slight increase in the use of nouns, accompanied by a corresponding decrease in pronouns. Prepositions and conjunctions remain stable after LLM simulation. On Wikipedia pages, the proportion of prepositions has steadily increased, while the proportions of other parts of speech have remained stable.

**Syllables**  Figures 22 and 23 illustrates that the proportion of one-syllable words declines in articles revised by LLMs, with Gemini employing even fewer such words. Meanwhile, the average syllables per word increase, suggesting a preference for polysyllabic words by LLMs. However, these two metrics remain relatively stable across different Wikipedia categories.

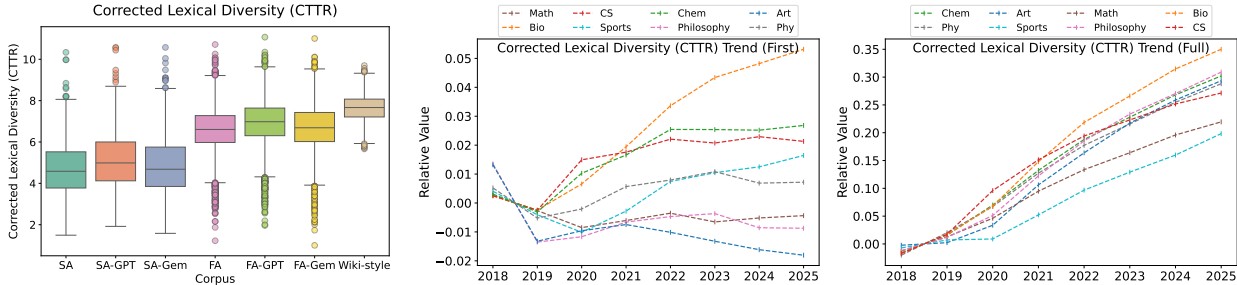

Figure 16: Corrected Type-Token Ratio (CTTR).

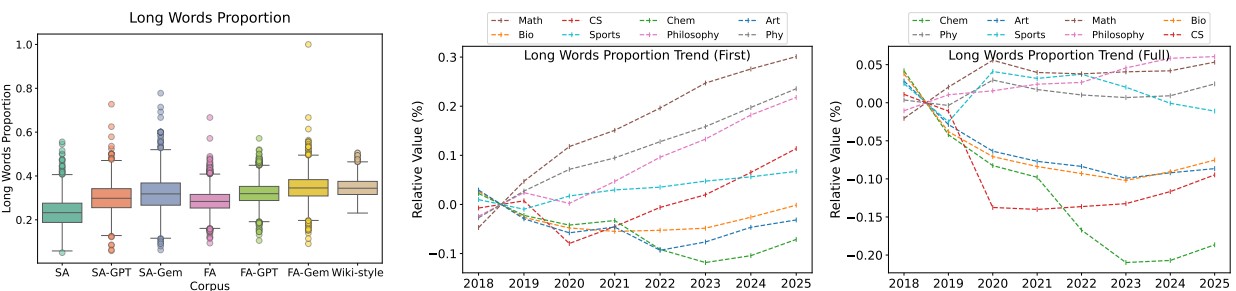

Figure 17: Long Words Rate.

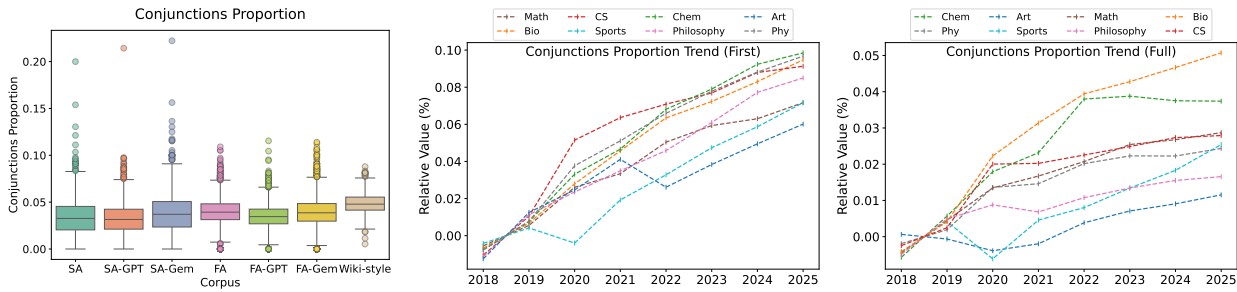

Figure 18: Conjunctions Proportion.

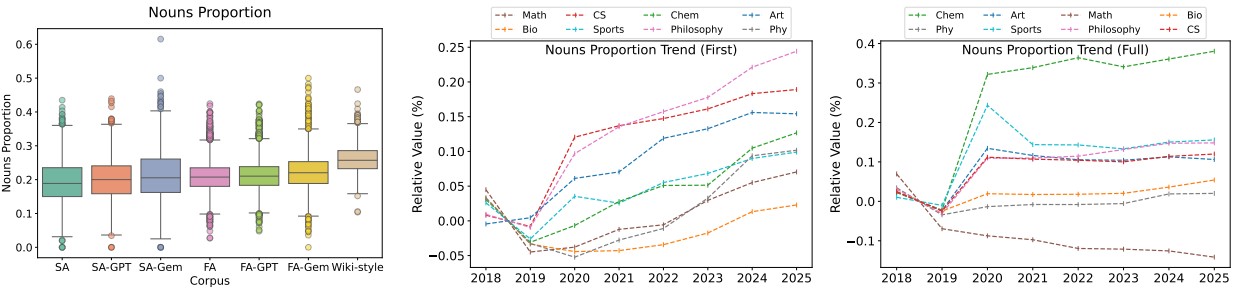

Figure 19: Nouns Proportion.

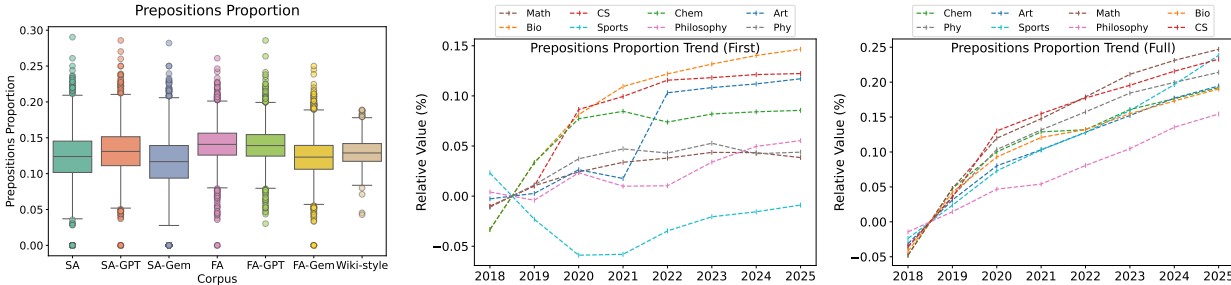

Figure 20: Prepositions Proportion.

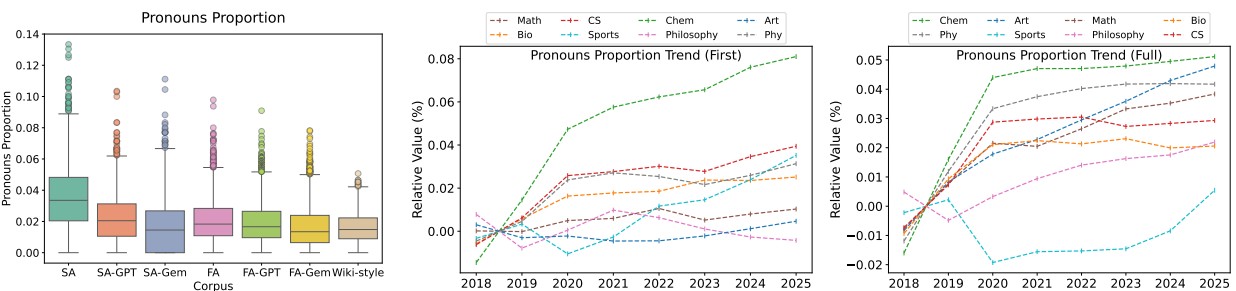

Figure 21: Pronouns Proportion.

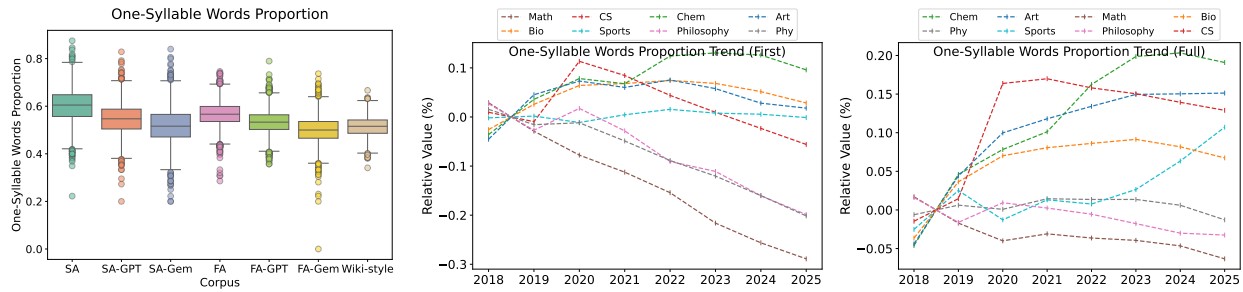

Figure 22: One-Syllable Words Proportion.

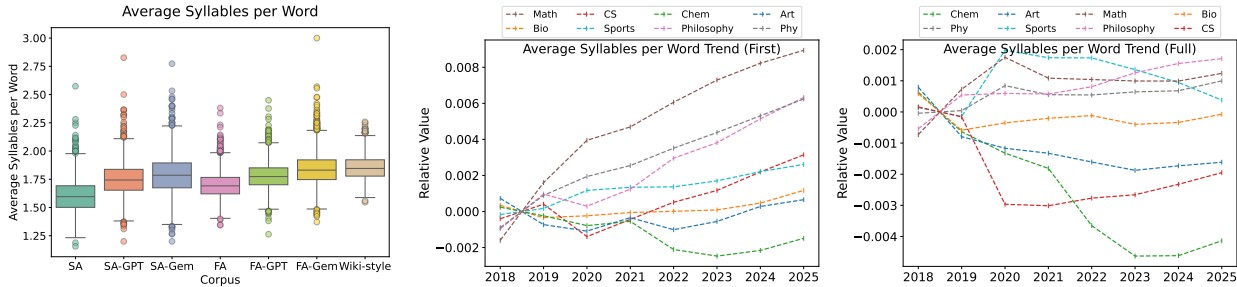

Figure 23: Average Syllables per Word.

## C.2   Sentence Level

**Sentence Length**   Figure 25 shows that both the average sentence length and the proportion of long sentences show a significant increase after being processed by the LLM. Additionally, the period from 2020 to 2025 has seen a notable rise in these two metrics across Wikipedia pages, indicating a trend towards longer sentence structures.

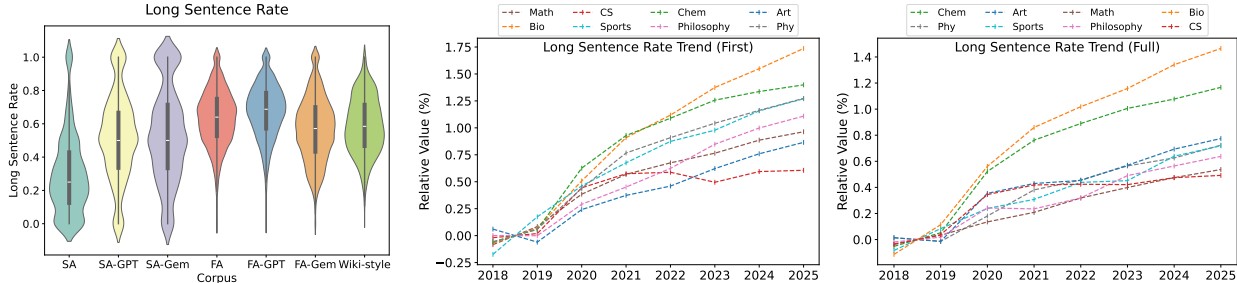

Figure 24: Long Sentence Rate.

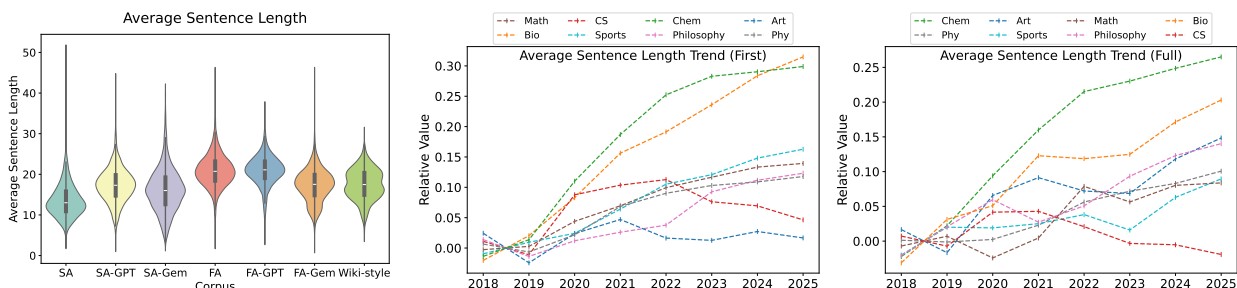

Figure 25: Average Sentence Length.

**Sentence Complexity**   According to Figures 26 and 27, after revisions by GPT, *Simple Articles* show an increase in complexity, while *Featured Articles* exhibit only minor changes. This may suggest that LLMs do not generate sentences at the highest possible complexity, but instead maintain complexity at a certain level. For real Wikipedia pages, a steady year-on-year increase in these two metrics has been observed, indicating a shift towards more complex sentence structures.

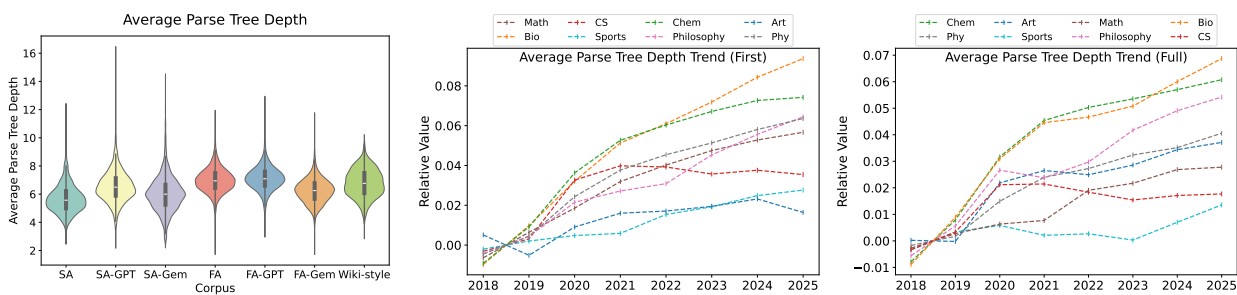

Figure 26: Average Parse Tree Depth.

**Pronoun and Article-Initial Sentences**   LLMs tend to avoid starting sentences with pronouns (*e.g.*, "It") or articles (*e.g.*, "The"), as shown in Figures 29 and 28. For example, it might replace *"The team*

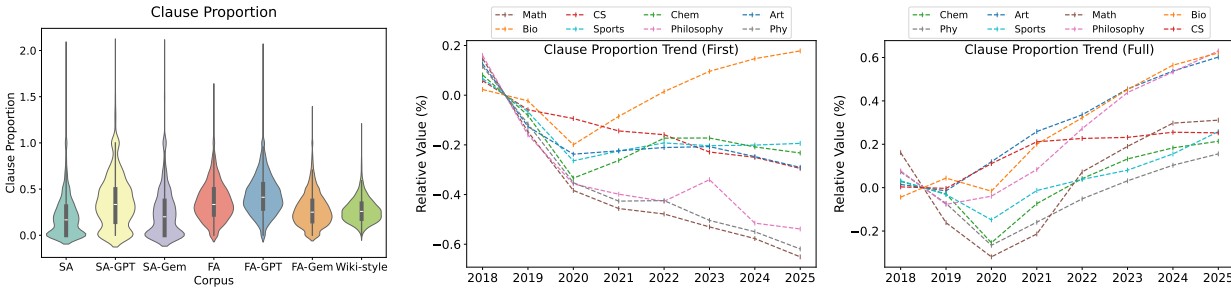

Figure 27: Clause Proportion

*worked hard to finish the project on time."* with *"Hard work from the team ensured the project was completed on time."* However, in real Wikipedia pages, Article-initial sentences have increased, while pronoun-initial sentences remain stable from 2020 to 2025.

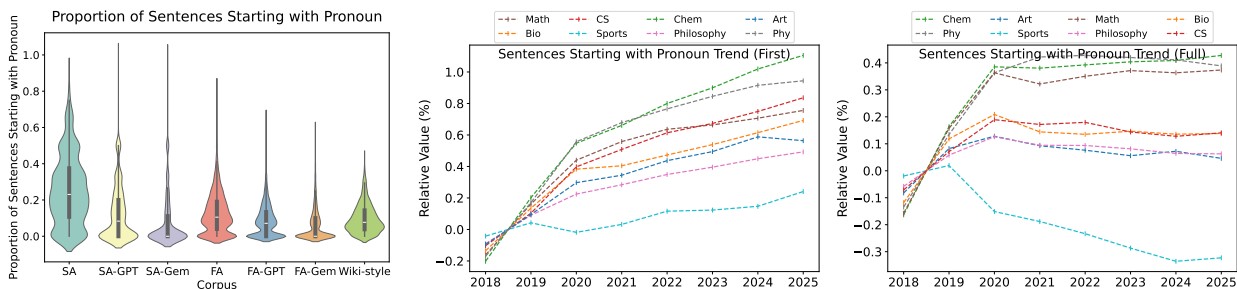

Figure 28: Proportion of Sentences Starting with Pronoun.

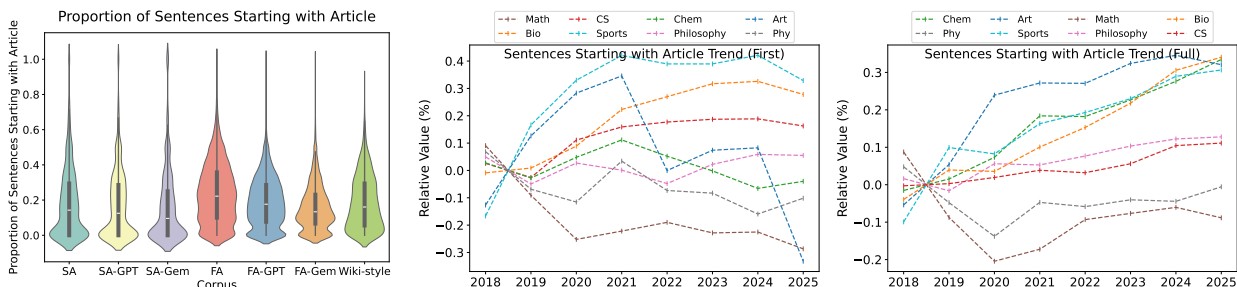

Figure 29: Proportion of Sentences Starting with Article.

## C.3  Paragraph Level

We use *Textstat*[10] to calculate six paragraph metrics. *Textstat* is an easy-to-use library to calculate statistics from the text. It provides a range of functions to analyze readability, sentence length, syllable count, and other important textual features. Through the LLM simulation process, we discover that LLMs tend to generate articles that are harder to read. Figure 30 suggests that the readability of Wikipedia pages has shown only slight variation over the years and does not appear to be influenced by LLMs at this stage.

**Dale-Chall Readability**: uses the concept of *difficult words*, combining it with the average sentence size to estimate readability.

---

[10]https://github.com/textstat/textstat

$$DC = 0.1579 * (\frac{difficultwords}{words} * 100) + 0.0496\frac{words}{sentences} \tag{6}$$

**Automated Readability Index**: Estimates readability by combining the average word length with the average sentence size.

$$ARI = 4.71\frac{characters}{words} + 0.5\frac{words}{sentences} - 21.43 \tag{7}$$

**Coleman-Liau Index**: Similarly to $ARI$, estimates readability by combining the average word length with the average sentence size.

$$CL = 5.88\frac{characters}{words} - 29.6\frac{sentences}{words} - 15.8 \tag{8}$$

**Flesch Reading Ease**: Using the average sentence size and amount of syllables per word, computes a value between 0 and 100, where 0 indicates the text is difficult to understand.

$$FRE = 206.835 - 1.015\frac{words}{sentences} - 84.6\frac{syllables}{words} \tag{9}$$

**Flesch-Kincaid Grade Level**: Same as $FRE$, but provides US grade levels instead of values between 0 and 100.

$$FK = 0.39\frac{words}{sentences} - 11.8\frac{syllables}{words} - 15.59 \tag{10}$$

**Gunning Fog Index**: Uses the concept of *complexwords*, which is the number of words with three or more syllables. The higher its value, the more difficult is the text to read.

$$GFI = 0.4(\frac{words}{sentences} + 100\frac{complexwords}{words}) \tag{11}$$

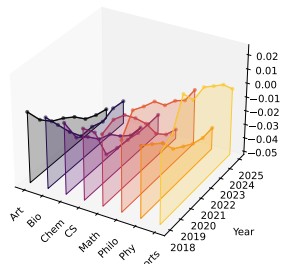

(a) Dale-Chall Readability.

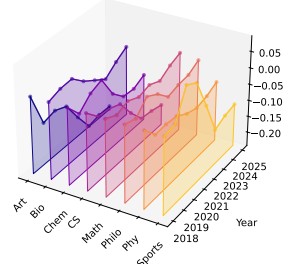

(b) Automated Readability.

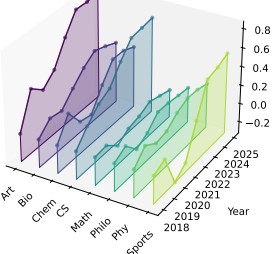

(c) Flesch Reading Ease.

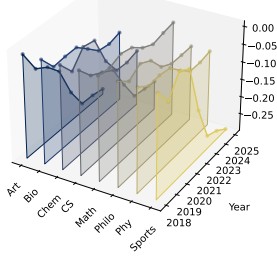

(d) Coleman-Liau Index.

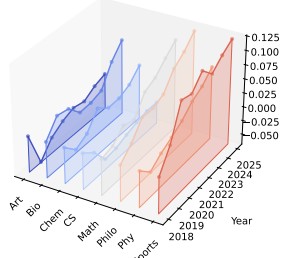

(e) Gunning Fog Index.

Figure 30: Changes in readability metrics of Wikipedia pages.

## D  Page Views

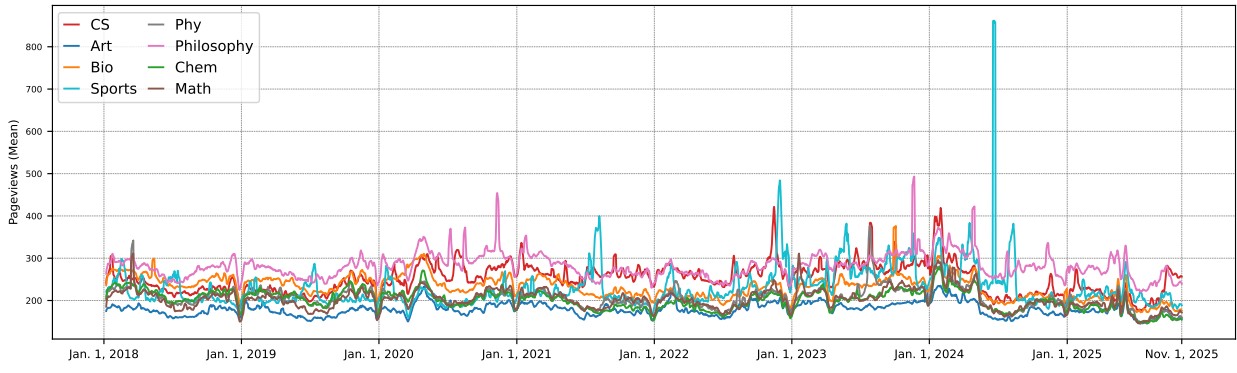

(a) Page views Across Different Categories in English Wikipedia.

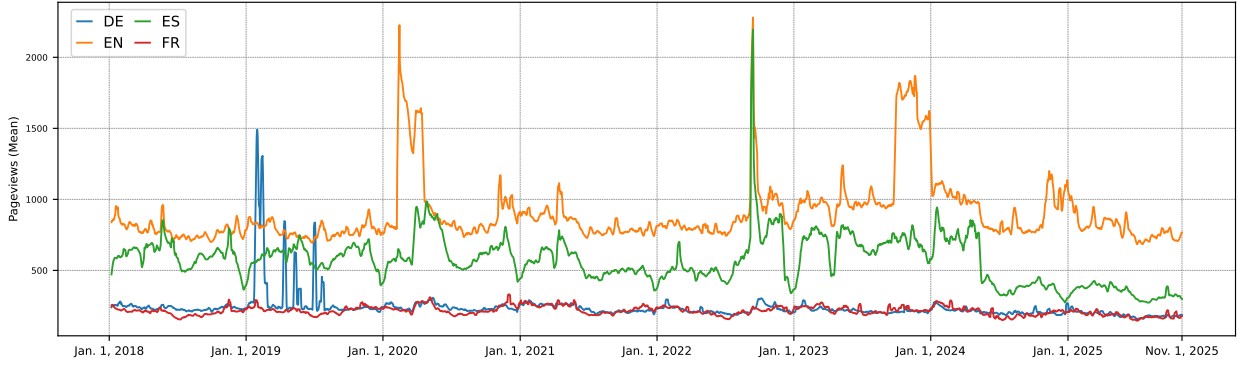

(b) Page views Across Wikipedia of Different Languages

Figure 31: Daily page views of Wikipedia pages. The y-axis represents page view values after smoothing with a seven-day time window and being transformed via mean aggregation.

## E  Machine Translation

These are the 12 languages in our benchmarks: *English* (eng-Latn-stan1293), *Modern Standard Arabic* (arb-Arab-stan1318), *Mandarin* (cmn-Hans-beij1234), *German* (deu-Latn-stan1295), *French* (fra-Latn-stan1290), *Hindi* (hin-Deva-hind1269), *Italian* (ita-Latn-ital1282), *Japanese* (jpn-Jpan-nucl1643), *Korean* (kor-Hang-kore1280), *Brazilian Portuguese* (por-Latn-braz1246), *Russian* (rus-Cyrl-russ1263), *Latin American Spanish* (spa-Latn-amer1254).

For *Google-T5* shown in Table 6, German (*DE*) initially has a *BLEU* score of 30.24, which rises to 44.18 in the GPT-processed benchmark, marking another substantial improvement.

We also evaluate our results using BERTScore, as shown in Tables 4, 5, and 7. Precision measures how many tokens in the candidate sentence are similar to tokens in the reference sentence, capturing how much of the candidate sentence is relevant to the reference. Recall measures how many tokens in the reference sentence are similar to tokens in the candidate sentence, capturing how much of the reference sentence is represented in the candidate. As for F1 Score, BERTScore combines precision and recall into an F1 score, the harmonic mean of the two. This balanced measure provides a single metric that reflects the accuracy and completeness of the candidate sentence relative to the reference.

Overall, our conclusion that LLM-influenced benchmarks inflate translation scores across different languages still holds when using BERTScore as the evaluation metric.

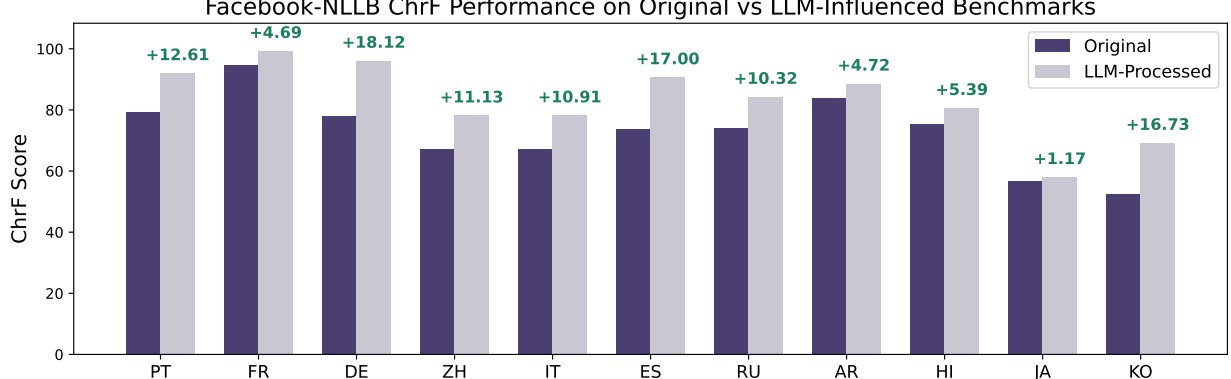

Figure 32: Facebook-NLLB ChrF scores on the original benchmark and the LLM-Influenced benchmark.

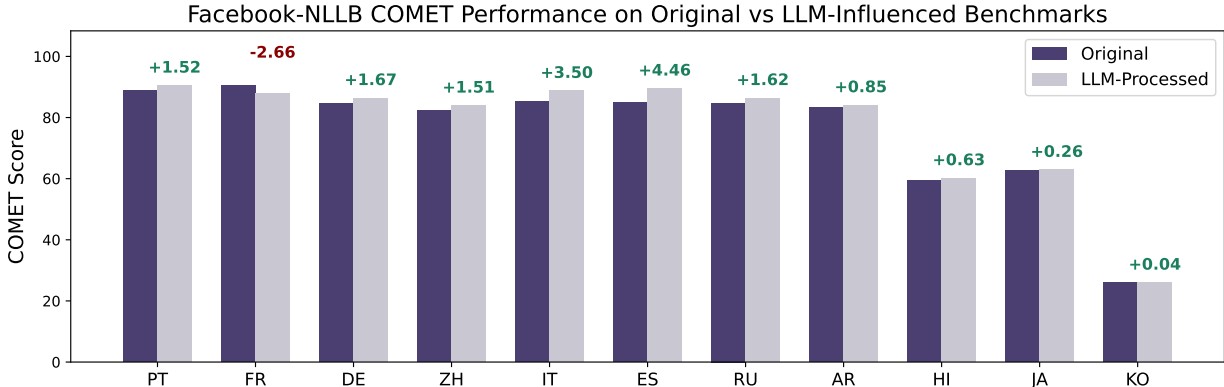

Figure 33: Facebook-NLLB COMET scores on the original benchmark and the LLM-Influenced benchmark.

| | Precision | | Recall | | F1 | |
|---|---|---|---|---|---|---|
| | O | G | O | G | O | G |
| AR | 0.869 | 0.891 | 0.854 | 0.880 | 0.861 | 0.886 |
| DE | 0.890 | 0.915 | 0.874 | 0.900 | 0.882 | 0.907 |
| ES | 0.885 | 0.953 | 0.867 | 0.943 | 0.876 | 0.948 |
| FR | 0.919 | 0.944 | 0.902 | 0.930 | 0.910 | 0.937 |
| HI | 0.876 | 0.900 | 0.865 | 0.892 | 0.870 | 0.896 |
| IT | 0.883 | 0.936 | 0.865 | 0.925 | 0.874 | 0.930 |
| JA | 0.829 | 0.850 | 0.808 | 0.824 | 0.818 | 0.836 |
| KO | 0.849 | 0.878 | 0.842 | 0.869 | 0.845 | 0.873 |
| PT | 0.923 | 0.946 | 0.913 | 0.937 | 0.918 | 0.941 |
| RU | 0.875 | 0.908 | 0.858 | 0.892 | 0.866 | 0.899 |
| ZH | 0.839 | 0.861 | 0.778 | 0.797 | 0.806 | 0.827 |

Table 4: BERTScore evaluation results on the Facebook-NLLB translation outputs.

| | Precision | | Recall | | F1 | |
|---|---|---|---|---|---|---|
| | O | G | O | G | O | G |
| AR | 0.861 | 0.872 | 0.854 | 0.869 | 0.857 | 0.870 |
| DE | 0.896 | 0.923 | 0.888 | 0.916 | 0.892 | 0.919 |
| ES | 0.884 | 0.952 | 0.871 | 0.949 | 0.877 | 0.951 |
| FR | 0.919 | 0.944 | 0.911 | 0.941 | 0.915 | 0.942 |
| HI | 0.812 | 0.822 | 0.785 | 0.798 | 0.798 | 0.809 |
| IT | 0.880 | 0.935 | 0.869 | 0.931 | 0.874 | 0.933 |
| JA | 0.608 | 0.612 | 0.625 | 0.629 | 0.617 | 0.620 |
| KO | 0.610 | 0.614 | 0.602 | 0.605 | 0.605 | 0.609 |
| PT | 0.929 | 0.954 | 0.926 | 0.951 | 0.927 | 0.952 |
| RU | 0.868 | 0.896 | 0.859 | 0.889 | 0.863 | 0.892 |
| ZH | 0.852 | 0.870 | 0.825 | 0.845 | 0.838 | 0.857 |

Table 5: BERTScore evaluation results on the Helsinki-NLP machine translation outputs.

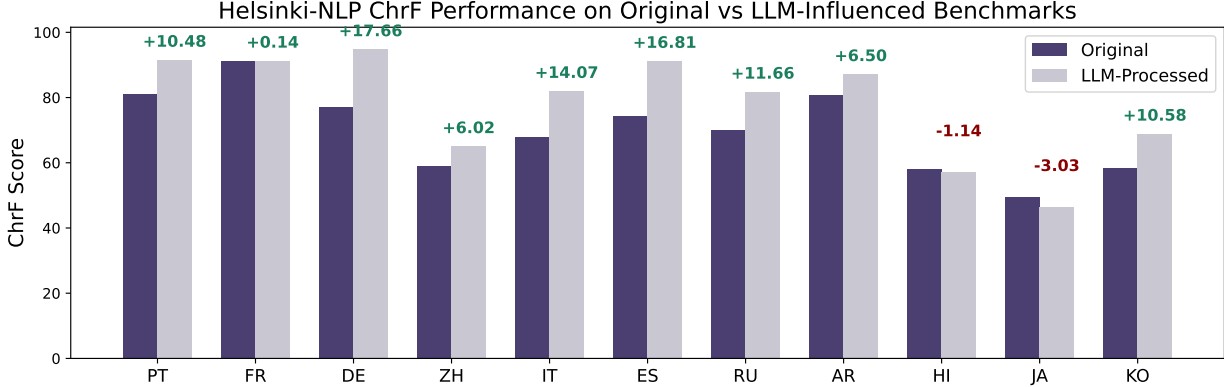

Figure 34: Helsinki-NLP ChrF scores on the original benchmark and the LLM-Influenced benchmark.

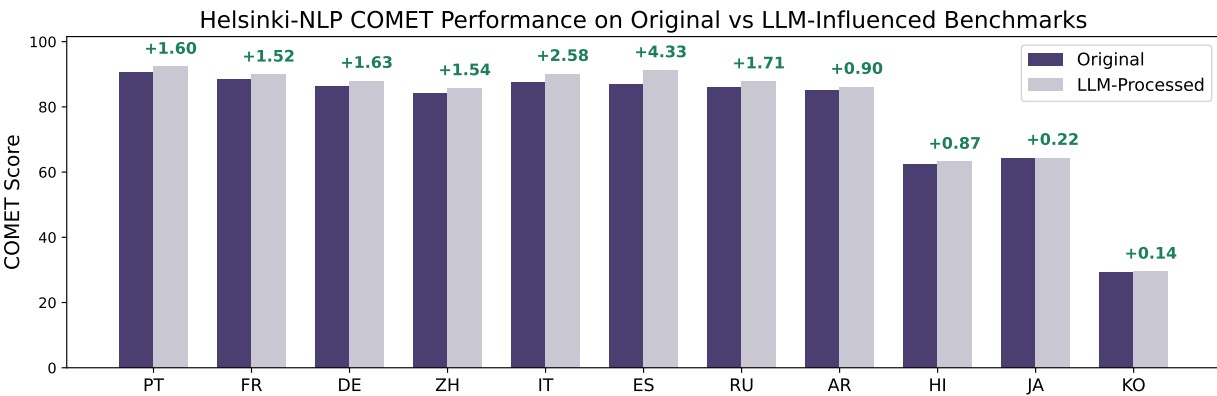

Figure 35: Helsinki-NLP COMET scores on the original benchmark and the LLM-Influenced benchmark.

# F  RAG

## F.1  Experiment Setup

Table 8 presents the LLM parameters employed in RAG simulations, such as the *knowledge cutoff date*, *temperature*, and *top-p*. Table 9 shows the annual number of questions generated by different LLMs.



**Prompt: Direct Asking**

```
prompt = (
    f"Answer the following questions.  The format should be as per 1.  C)..."
    f"Need to answer all questions and mark the question number."
    f"Only need to give each answer without explanation.  Questions:  {text}"
    f"The format should be as per 1.  C)...\n2.  C)..."
    f"All questions are required to be answered.  Don't skip any."
)
```



Figure 36: Prompt used in the direct asking setting.

|    | BLEU | | ChrF | | COMET | |
|----|-------|-------|-------|-------|-------|-------|
|    | O | G | O | G | O | G |
| DE | 71.52 | 80.09 | 84.27 | 93.62 | 83.91 | 85.63 |
| FR | 68.33 | 65.93 | 87.86 | 86.32 | 85.49 | 87.01 |

Table 6: Google-T5 results on BLEU, ChrF, and COMET metrics.

|    | Precision | | Recall | | F1 | |
|----|-------|-------|-------|-------|-------|-------|
|    | O | G | O | G | O | G |
| DE | 0.873 | 0.898 | 0.845 | 0.869 | 0.858 | 0.883 |
| FR | 0.887 | 0.907 | 0.849 | 0.869 | 0.867 | 0.887 |

Table 7: BERTScore evaluation results (Precision, Recall, F1) on the Google-T5 translation outputs.

| Models | Knowledge Cutoff | Temperature | Top-p |
|--------|------------------|-------------|-------|
| GPT-3.5 | September 2021 | 1.0 | 1.0 |
| GPT-4o-mini | October 2023 | 1.0 | 1.0 |
| Gemini-1.5-flash | May 2024 | 1.0 | 0.95 |
| DeepSeek-V3 | July 2024 | 1.0 | 1.0 |
| Gemini 3 Pro | January 2025 | 1.0 | 1.0 |

Table 8: LLM parameters Used in RAG simulations.

| Year | 2020 | 2021 | 2022 | 2023 | 2024 |
|------|------|------|------|------|------|
| Number of GPT generated Questions | 348 | 453 | 390 | 426 | 240 |
| Number of Gemini generated Questions | 348 | 453 | 393 | 426 | 240 |

Table 9: Annual Number of Questions Generated by Different LLMs.

**Prompt: Full Texts Provided**

```
prompt = (
    f"Use context to answer user questions."
    f"question:  {question}"
    f"Reference context:  {content}"
    f"Only need to give the correct option without explanation.  Don't miss ')' or
option."
    f"If there is no answer in the content, just return None.  Don't give a
string."
)
```

Figure 37: Prompt used in the full texts provided setting.

### F.2 Detailed Results

Tables 10, 11, 12, 13, 14, and 15 present detailed RAG results. Questions generated from Wikinews in 2024 are likely the most up-to-date; therefore, we focus on their results, which correspond to the last row in each table. The results indicate that LLM-revised content tends to be less effective as a knowledge source, as accuracy based on LLM-revised texts is often lower than that based on the original texts.

| Y | Direct | RAG | R (GPT) | R (Gem1.5) | **RAG (Gem3)** | Full (Original) | Full (GPT) | Full (Gem1.5) | **Full (Gem3)** |
|----|--------|--------|---------|------------|----------------|-----------------|------------|---------------|-----------------|
| 20 | 75.86% | 85.34% | 85.63% | 79.60% | **85.63%** | 95.98% | 95.40% | 87.36% | **94.25%** |
| 21 | 71.74% | 86.31% | 88.96% | 79.69% | **82.56%** | 96.03% | 96.03% | 88.08% | **93.38%** |
| 22 | 80.00% | 89.49% | 87.18% | 84.10% | **85.90%** | 95.64% | 95.64% | 88.97% | **93.85%** |
| 23 | 77.46% | 87.09% | 87.09% | 83.33% | **84.98%** | 96.01% | 94.84% | 87.09% | **93.19%** |
| 24 | 66.67% | 83.33% | 84.58% | 82.08% | **83.33%** | 95.83% | 95.83% | 88.75% | **90.42%** |

Table 10: GPT-4o-mini performance on RAG task (problem generated by GPT).

| Year | Direct Ask | RAG | RAG (GPT) | RAG (Gem) | Full (Original) | Full (GPT) | Full (Gem) |
|------|-----------|--------|-----------|-----------|-----------------|------------|------------|
| 2020 | 66.95% | 82.76% | 82.47% | 75.86% | 93.68% | 91.38% | 84.20% |
| 2021 | 64.68% | 81.90% | 82.34% | 75.06% | 94.04% | 93.82% | 82.12% |
| 2022 | 73.54% | 86.01% | 85.75% | 78.88% | 94.66% | 93.89% | 83.21% |
| 2023 | 69.95% | 82.39% | 83.10% | 78.40% | 92.49% | 92.25% | 83.57% |
| 2024 | 61.25% | 79.58% | 75.42% | 75.42% | 92.92% | 92.92% | 82.92% |

Table 11: GPT-4o-mini performance on RAG task (problem generated by Gemini).

| Year | Direct Ask | RAG | RAG (GPT) | RAG (Gem) | Full (Original) | Full (GPT) | Full (Gem) |
|------|-----------|--------|-----------|-----------|-----------------|------------|------------|
| 2020 | 68.68% | 77.59% | 78.16% | 74.14% | 86.21% | 87.93% | 87.36% |
| 2021 | 67.11% | 79.25% | 79.25% | 74.17% | 87.42% | 88.30% | 84.99% |
| 2022 | 70.26% | 82.82% | 80.77% | 78.97% | 88.46% | 90.51% | 88.46% |
| 2023 | 64.08% | 74.88% | 76.06% | 71.83% | 86.85% | 88.73% | 84.27% |
| 2024 | 60.42% | 77.92% | 75.83% | 75.83% | 92.08% | 89.17% | 83.75% |

Table 12: GPT-3.5 Performance on RAG task (problem generated by GPT).

| Year | Direct Ask | RAG | RAG (GPT) | RAG (Gem) | Full (Original) | Full (GPT) | Full (Gem) |
|------|-----------|-----|-----------|-----------|-----------------|------------|------------|
| 2020 | 66.95% | 72.70% | 72.41% | 68.97% | 77.87% | 79.31% | 77.59% |
| 2021 | 58.72% | 73.73% | 71.74% | 68.21% | 81.02% | 79.47% | 74.17% |
| 2022 | 62.09% | 74.05% | 72.77% | 69.47% | 82.44% | 82.19% | 80.41% |
| 2023 | 56.57% | 73.24% | 74.88% | 67.14% | 77.46% | 79.58% | 74.65% |
| 2024 | 55.00% | 71.67% | 70.00% | 65.00% | 77.92% | 80.42% | 76.67% |

Table 13: GPT-3.5 Performance on RAG task (problem generated by Gemini).

| Year | Direct Ask | RAG | RAG (GPT) | RAG (Gem) | Full (Original) | Full (GPT) | Full (Gem) |
|------|-----------|-----|-----------|-----------|-----------------|------------|------------|
| 2020 | 89.66% | 81.03% | 81.90% | 78.45% | 98.28% | 97.70% | 90.52% |
| 2021 | 84.55% | 77.26% | 79.25% | 69.09% | 97.57% | 97.79% | 87.20% |
| 2022 | 90.00% | 80.77% | 81.54% | 75.90% | 97.69% | 97.44% | 90.00% |
| 2023 | 83.57% | 73.00% | 76.29% | 69.72% | 96.71% | 95.54% | 88.03% |
| 2024 | 82.08% | 75.42% | 72.50% | 75.42% | 97.08% | 96.67% | 86.25% |

Table 14: DeepSeek-V3 Performance on RAG task (problem generated by GPT).

| Year | Direct Ask | RAG | RAG (GPT) | RAG (Gem) | Full (Original) | Full (GPT) | Full (Gem) |
|------|-----------|-----|-----------|-----------|-----------------|------------|------------|
| 2020 | 83.62% | 74.14% | 75.57% | 69.54% | 94.54% | 95.11% | 83.05% |
| 2021 | 54.97% | 69.98% | 72.41% | 63.36% | 95.81% | 94.70% | 81.68% |
| 2022 | 84.48% | 78.12% | 78.37% | 65.39% | 96.18% | 94.91% | 84.99% |
| 2023 | 65.02% | 73.00% | 74.88% | 64.55% | 95.54% | 94.37% | 83.33% |
| 2024 | 75.83% | 74.17% | 70.42% | 70.00% | 95.42% | 93.75% | 84.58% |

Table 15: DeepSeek-V3 Performance on RAG task (problem generated by Gemini).

### F.3 Case Study

---

**Example 1 - Keyword Replacement**

**Title:** NASA says object that hit Florida home is from International Space Station[a]

**Question:** On which date did NASA release a pallet containing old nickel–hydride batteries from the International Space Station?
**A)** March 8, 2021 **B)** March 11, 2021 **C)** April 22, 2024 **D)** March 8, 2020

**Original Context:** . . . A pallet containing old nickel–hydride batteries was released from the ISS on **March 11, 2021**, after new batteries were installed. . . .

**LLM Revised Context:** . . . The debris, part of a 5,800-lb cargo pallet released from the ISS in **March 2021**, unexpectedly survived atmospheric re-entry. . . .

---

[a]https://en.wikinews.org/wiki/NASA_says_object_that_hit_Florida_home_is_from_International_Space_Station

---

Figure 38: The news revised by LLMs omits key information about the specific date that NASA released the pallet, causing the RAG system unable to determine the correct date and ultimately selecting **A**.

---

**Example 2 - Keyword Replacement**

**Title:** Latin American expedition of Viktor Pinchuk: meeting with the traveler took place in Yalta[a]

**Question:** What hobby involves collecting recordings of ethnic performers and is practiced by Viktor Pinchuk?
**A)** Philophony **B)** Ethnomusicology **C)** Hobo tourism **D)** Cultural preservation

**Original Context:** . . . From every trip or an expedition, Viktor Pinchuk brings CDs with recordings of ethnic performers; the traveler's collection has already accumulated a significant number of them (not counting several hundred digital editions of world-famous musicians). The hobby is called **"philophony"**, and the subject of it is called a philophonist. . . .

**LLM Revised Context:** . . . Pinchuk, a self-described **"philophonist,"** has amassed hundreds of CDs and digital recordings of ethnic and world music. . . .

---

[a]https://en.wikinews.org/wiki/Latin_American_expedition_of_Viktor_Pinchuk:_meeting_with_the_traveler_took_place_in_Yalta

---

Figure 39: The RAG system mistakenly selects B when using the LLM-revised text because the revision omits key details, such as the explicit mention of the hobby's name, "philophony."

---

**Example 3 - Keyword Replacement**

**Title:** New Zealand Navy ship HMNZS Manawanui capsizes one nautical mile from shore[a]
**Question:** What was the name of the Royal New Zealand Air Force aircraft that assisted in the evacuation of the crew from HMNZS Manawanui?
**A)** Boeing P-8 Poseidon **B)** Airbus A320 **C)** Lockheed Martin C-130J **D)** Boeing 737,
**Original Context:** . . . They were rescued with assistance from the Rescue Coordination Centre (RCCNZ) and **a Royal New Zealand Airforce Boeing P-8 Poseidon**. . . .
**LLM Revised Context:** . . . All 75 crew were safely evacuated with assistance from the Rescue Coordination Centre and the **Royal New Zealand Air Force**.

---
[a]`https://en.wikinews.org/wiki/New_Zealand_Navy_ship_HMNZS_Manawanui_capsizes_one_nautical_mile_from_shore`

Figure 40: LLMs omit key information, such as the aircraft's name.

---

**Example 4 - Abbreviation Ambiguity Misleading**

**Title:** At least 20 die in Odesa in Russian missile strike, Ukraine reports[a]
**Question:** How many employees of the State Emergency Service of Ukraine were reported as victims of the missile strikes in Odesa?
**A)** One **B)** Five **C)** Seven **D)** Ten
**Original Context:** . . . Among the dead are an employee of the State Service of Ukraine for Emergency Situations (SSES) and a paramedic. . . . Among the victims are **seven employees of the State Emergency Service.** . . .
**LLM Revised Context:** . . . Among the deceased are a staff member of the State Service of Ukraine for Emergency Situations (SSES) and a paramedic. . . . **Seven SSES personnel** were among the injured, and medical workers also sustained injuries. . . .

---
[a]`https://en.wikinews.org/wiki/At_least_20_die_in_Odesa_in_Russian_missile_strike,_Ukraine_reports`

Figure 41: The original text use the full name "*seven employees of the State Emergency Service*," while the LLM's revised text abbreviated this to "*seven SSES personnel*," causing RAG to incorrectly choose **A**.

---

**Example 5 - Introduction of Modifiers**

**Title:** Arizona bans abortion for genetic abnormalities[a]
**Question:** What does Senate Bill 1457 in Arizona classify as a Class 6 felony?
**A)** Seeking or performing an abortion because of a severe fetal abnormality
**B)** Seeking or performing an abortion due to the presence of a genetic abnormality in the child
**C)** Distributing abortion-inducing drugs via courier **D)** Soliciting funds for an abortion
**Original Context:** . . . The bill makes it a Class 6 felony, the least severe, to seek or perform an abortion **"because of a genetic abnormality of the child"**, defined as "the presence or presumed presence of an abnormal gene expression in an unborn child," but not a "severe fetal abnormality" considered "incompatible with life." . . .
**LLM Revised Context:** . . . Arizona Governor Doug Ducey signed Senate Bill 1457 into law on Tuesday, effectively **banning abortions sought solely due to fetal genetic abnormalities**. The bill, which passed the Republican-controlled legislature after twice stalling and undergoing amendments to secure necessary votes, classifies seeking or performing such abortions as a Class 6 felony.

---
[a]`https://en.wikinews.org/wiki/Arizona_bans_abortion_for_genetic_abnormalities`

Figure 42: Although both versions explicitly exclude "severe fetal abnormalities," the revision changes "genetic abnormality" to "fetal genetic abnormalities," causing LLMs to misinterpret the information.

