# OpenReview forum: "Wikipedia in the Era of LLMs: Evolution and Risks"
_TMLR — Accepted by TMLR_

### Review · Reviewer_dwkh · 2025-11-28

**Summary Of Contributions:**

This paper studies the impact of LLMs on Wikipedia, and they measure direct and indirect impacts of LLMs in their work. Based on their experiments, the authors conclude that LLMs do have some impact on Wikipedia but it is overall not significant/uncertain. They also believe that some machine translation model scores could be inflated due to LLMs' influence on Wikipedia.

Some key strengths of the paper include:
- a principled experimental design and really nice problem to study
- clear organization of the paper
- inclusion of impacts for practical tasks like machine translation and RAG pipelines
- i also really liked the case study at the end, and it was arguably the most clear and impactful section to me in the paper

Some key weaknesses of the paper are:
- although the existing experimental setup is technically sound for that they are doing, they do not truly capture what the authors originally set out to measure
- the main text is also not self-complete. there are many key details missing from protocol descriptions that make it hard for a reader to follow along smoothly
- the direct impact analysis section is arguably weak and for the strong claims made in that section, the experiments need to be studied more thoroughly as well

**Additional Comments:**

- I don't quite understand what is $f_{i}^{*}(S)$ even after reading the text. I am also confused as to how can you determine if some text is affected by an LLM based on the formula. What is $\eta^{\hat}(S)$ in Eq 1 supposed to capture, intuitively speaking and how does the interaction between different terms capture that?
- Why is there no $i$ anywhere in $f(S_2)$ or $f(S_1)$ in Eq 2?
- I don't quite understand the LLM Simulation setting, we have GPT-4o-mini and Gemini-1.5-Flash, but it is not clear from the main text what are they exactly simulating or benchmarked on. I took a quick look at the appendix but there's a lot going on there. It would really help if you could clarify the simulation setting more clearly in the main text.
- The results section is also a bit hard to follow. In the sentence "Although we have plotted...from 2020 in **these** figures" do they mean the figures in the main text or the appendix? Following that, in "...the trends summarized in **the** table are based on..." which table does the highlighted "the" refer to? Table 1 caption does not clarify this either.
- The term "page view" is a little ambiguous in this context as it can mean the number of views for a given wikipedia page (the intended meaning) or it could mean the rendered view of a Wikipedia page (which is how it reads like at the beginning).
- What does the "Full Content" setting mean in Figure 8? The input is the question+the wiki article that was used to construct those questions? Those setting names from the figure should be referenced in the main text.
- Where do the chatGPT processed articles come into picture? It abruptly appears the questioning methods paragraph and the exact protocol on how it is was used is not clear.
- Results of the case study are really good.
- The authors have more than 1 page to make the main text more complete without changing the category of the submission, and I recommend utilizing it as I felt a lot of disconnections when reading the paper as some key details that would've helped me understand the paper better were absent.

**Audience:**

Yes

**Audience Explanation:**

The impact of LLMs of existing bodies of text like Wikipedia and to extend it to the whole internet is a meaningful research problem as it also directly guides many pretraining/posttraining decisions for frontier models down the line.

**Broader Impact Concerns:**

There are no broader impact concerns.

**Claims And Evidence:**

No

**Claims Explanation:**

- When the authors say "crucial" and "additionally" are favored by LLMs, it is missing a citation. Figure 2 shows an increase in their frequencies but is it enough to establish a direct causal link between that LLMs? Consequently, the Impact metric computed is also based on the assumption that crucial and additionally are favored by LLMs (which may be true but I don't see enough evidence of that being true). Instead of word frequencies, I think perplexity would be a much more reliable metric to measure here as we *know* LLMs are trained to achieve lower perplexity. Another thing to do could be to ask LLMs to rewrite pre-LLM articles and plot the before and after word frequency trends (you don't need to do it for all the pre-LLM articles but sufficiently big and diverse sample should be enough, and using open LLMs might further help reduce the compute cost).

- For 4.1, I am not convinced that the simulation experiment of rewriting a sentence with an LLM truly captures the LLM impact as the instruction to revise does not include any incentive for the LLM to actually replace some words. It can high some biases for sure, but it is not representative of what the authors are trying to measure. A much more representative task would be ask it to write some text from scratch on the same topic based on the same reference content and then compare the two versions (maybe you already use this data for indirect impact analysis, but I am not sure if you restrict the LLM to use the same references as the original article there).

- For the linguistic style analysis, it is good to see different linguistic criteria but it is not clear from the text which ones have been proven to be preferred by humans. For instance, I would guess LLMs prefer more active voice than passive voice but has this been shown or studied before? Same argument for the remaining criteria. If they have, please clearly state them in the main text.

- I am also not sure we can establish a direct causal connection between higher performance on the GPT-processed benchmark vs the original benchmark and the fact that this suggests that LLMs can inflate translation scores. First, GPT-4o-mini $\ne$ all LLMs, and it is a pretty weak model. Second, in addition to those quantitative numbers, we should try to understand the failure cases in original v/s the success cases in the GPT-processed benchmark (where do GPT translations differ and how correct are they). Because even if GPT constructs incorrect translations, I don't understand how that would strictly boost a translation model's score (I think it's just as likely to drop). I am really curious to understand this.

**Requested Changes:**

- Please consider presenting Tables 2 and 3 visually. It is really hard to tally through > 100 numbers in a table. A simple bar chart on the other hand would've instantly told me the story (just an example, there may a better way to represent the data than this). This is a minor change though and it won't impact my decision
- I also find the models used out of date, and while I understand that you cannot possibly study the impact of all the LLMs, include some good models from the recent few months (and preferably open models) would really help generalize your findings. Looking at the analysis for a more recent model's data would help me make a better decision for the paper.
- Please address the concerns I have raised regarding the representation of your experimental setup as it would strongly impact my decision for the submission
- I have also compiled some other cosmetic comments that would help improve the delivery of the paper.

---

> ### Author Response · Authors · 2025-12-16
> **Author Response 1 to Reviewer dwkh**
>
> Thank you for your questions and suggestions. We hope our response addresses your concerns.
>
> **Q1.1:** When the authors say "crucial" and "additionally" are favored by LLMs, it is missing a citation. Figure 2 shows an increase in their frequencies but is it enough to establish a direct causal link between that and LLMs? Consequently, the Impact metric computed is also based on the assumption that crucial and additionally are favored by LLMs (which may be true but I don't see enough evidence of that being true).
>
> **A1.1:** This observation is actually supported by some studies cited in the other places of our paper. We have added more details after this argument, which identifies “crucial” and “additionally” as words favored by LLMs and increasingly used by humans after the emergence of LLMs.
>
> ---
>
> **Q1.2:** Instead of word frequencies, I think perplexity would be a much more reliable metric to measure here as we know LLMs are trained to achieve lower perplexity.
>
> **A1.2:** We agree that perplexity is a good metric and has been widely used in detecting LLM-generated content. However, our goal is to measure the evolution of Wikipedia pages rather than to detect whether a specific page was produced by an LLM. We also have to admit that computing perplexity can be time-consuming given our large dataset. In addition, incomplete data cleaning, such as including some random html symbols, can cause great bias in perplexity calculation. We do not guarantee that our dataset is fully clean plain text without any errors. Overall, word-frequency based methods are better for our research focus and also easier to interpret.
>
> ---
>
> **Q1.3:** Another thing to do could be to ask LLMs to rewrite pre-LLM articles and plot the before and after word frequency trends (you don't need to do it for all the pre-LLM articles but sufficiently big and diverse sample should be enough, and using open LLMs might further help reduce the compute cost).
>
> **A1.3:** We have added a plot in Figure 2 to show the word frequency of “crucial” and “additionally” before and after LLM processing. If you have any other comparison results you’d like to know, we would be happy to provide them.
>
> ---
>
> **Q2:** For 4.1, I am not convinced that the simulation experiment of rewriting a sentence with an LLM truly captures the LLM impact as the instruction to revise does not include any incentive for the LLM to actually replace some words. It can high some biases for sure, but it is not representative of what the authors are trying to measure. A much more representative task would be ask it to write some text from scratch on the same topic based on the same reference content and then compare the two versions (maybe you already use this data for indirect impact analysis, but I am not sure if you restrict the LLM to use the same references as the original article there).
>
> **A2:** Thank you for your suggestion. But considering the following reasons, our original setup may be more appropriate:
> (1) In this part, our focus is the use of language (more precisely, the use of words), rather than the generation of content.
> (2) The variability in generated content is too high, as it is easily influenced by prompts. Therefore, it is difficult to use it to quantitatively estimate the impact of LLMs.
> (3) Our original approach is easy to implement and aligns with real-world scenarios.
>
> ---
>
> **Q3:** For the linguistic style analysis, it is good to see different linguistic criteria but it is not clear from the text which ones have been proven to be preferred by humans. For instance, I would guess LLMs prefer more active voice than passive voice but has this been shown or studied before? Same argument for the remaining criteria. If they have, please clearly state them in the main text.
>
> **A3:** Thank you for your suggestion. We have added the following citations in the new version.
> (1) Herbold et al. (2023)  revealed that the **lexical diversity** of humans is higher than that of ChatGPT-3 but lower than that of ChatGPT-4, suggesting newer models have surpassed human writing in that metric.
> (2) Georgiou (2025) showed that LLM-generated text employed more **nouns,** while human-written text employed more **auxiliaries** and **pronouns**.
> (3) Reinhart et al. (2025) revealed that LLMs use present participial **clauses** at 2 to 5 times the rate of human text while use the **passive voice** at roughly half the rate as human texts.
>
> ---

---

> > ### Author Response · Authors · 2025-12-16
> > **Author Response 2 to Reviewer dwkh**
> >
> > **Q4:** I am also not sure we can establish a direct causal connection between higher performance on the GPT-processed benchmark vs the original benchmark and the fact that this suggests that LLMs can inflate translation scores. First, GPT-4o-mini ≠ all LLMs, and it is a pretty weak model. Second, in addition to those quantitative numbers, we should try to understand the failure cases in original v/s the success cases in the GPT-processed benchmark (where do GPT translations differ and how correct are they). Because even if GPT constructs incorrect translations, I don't understand how that would strictly boost a translation model's score (I think it's just as likely to drop). I am really curious to understand this.
> >
> > **A4:** We will now address your questions regarding machine translation in turn:
> >
> > (1) First, you may have misunderstood our conclusions. Our comparison of model translation performance on benchmarks before and after LLM processing was not to demonstrate that "LLMs can inflate translation scores" as you mentioned. On the contrary, we were not exploring how to improve machine translation performance, but rather investigating the risks of LLM contamination of benchmarks. Therefore, regarding the phenomenon of "better performance on benchmarks after GPT processing than on the original benchmarks," we want to show that because the benchmarks are affected, our actual evaluation of model performance may be "artificially inflated"—the model performance is not actually that good, but rather due to the contamination of the benchmarks. This is the risk aspect mentioned in our article.
> >
> > (2) We need to point out that existing machine translation metrics (such as BLEU, chrf, comet, etc. used in our article) are more based on similarity and are a comprehensive comparison with the benchmark and the original text, rather than simply judging the accuracy of the translation. When scoring, the distinction is between "good" and "bad," not between "right" and "wrong." Understanding this, I think this phenomenon is very intuitive: the benchmark is AI, the evaluation is also AI, and preferences are similar. AI will naturally tend to give higher scores to its own peers, and the similarity to human-annotated results (i.e., origin bench) will certainly not be as high, even if it is the correct answer.
> >
> > (3) Regarding the case study you mentioned, the reason we didn't discuss it in detail in the text is because it's difficult to judge the translation results from a human perspective, as this requires readers to understand a language other than English. We can give a few examples and the full results are in our github repositories.
> >
> > | id | Model Translation | Origin Benchmark | LLM Benchmark |
> > | ----- | ----- | ----- | ----- |
> > | 0 | "Nous avons maintenant des souris de 4 mois qui ne sont pas diabétiques et qui étaient auparavant diabétiques", a- t- il ajouté.  | « Nous avons à présent des souris de 4 mois qui ne sont pas diabétiques alors qu'elles l'étaient auparavant », a-t-il ajouté.  | « Nous avons maintenant des souris de 4 mois qui ne sont pas diabétiques et qui étaient auparavant diabétiques », a-t-il ajouté.  |
> > | 1 | Le Dr Ehud Ur, professeur de médecine à l'Université Dalhousie à Halifax, en Nouvelle-Écosse, et président de la division clinique et scientifique de l'Association canadienne du diabète, a averti que la recherche est encore à ses débuts.  | Le Dr Ehud Ur, professeur de médecine à l'Université Dalhousie de Halifax (Nouvelle-Écosse) et président de la division clinique et scientifique de l'Association canadienne du diabète, a averti que la recherche en était encore à ses débuts.  | Le Dr Ehud Ur, professeur de médecine à l'Université Dalhousie à Halifax, en Nouvelle-Écosse, et président de la division clinique et scientifique de l'Association canadienne du diabète, a averti que la recherche en est encore à ses débuts.  |
> > | 2 | Comme certains autres experts, il est sceptique quant à savoir si le diabète peut être guéri, notant que ces résultats n'ont aucune pertinence pour les personnes qui ont déjà le diabète de type 1\.  | À l'instar d'autres experts, il se montre sceptique quant à la possibilité de guérir le diabète, faisant remarquer que ces résultats ne sont pas applicables aux personnes qui souffrent déjà de diabète de type 1\.  | Comme certains autres experts, il est sceptique quant à la possibilité de guérir le diabète, notant que ces découvertes n'ont aucune pertinence pour les personnes qui ont déjà le diabète de type 1\.  |
> >
> > ---

---

> > > ### Author Response · Authors · 2025-12-16
> > > **Author Response 3 to Reviewer dwkh**
> > >
> > > **RQ1:** Please consider presenting Tables 2 and 3 visually. It is really hard to tally through \> 100 numbers in a table. A simple bar chart on the other hand would've instantly told me the story (just an example, there may be a better way to represent the data than this). This is a minor change though and it won't impact my decision.
> > >
> > > **A1:** Thank you for the helpful suggestion. In the revised version, we have replaced these tables with clear figures. Specifically, for each model and each metric BLEU, ChrF, COMET, we now provide bar charts comparing the original benchmark and the GPT-processed benchmark (G), with Δ \= G − O annotated above each bar.
> > >
> > > ---
> > >
> > > **RQ2:** I also find the models used out of date, and while I understand that you cannot possibly study the impact of all the LLMs, including some good models from the recent few months (and preferably open models) would really help generalize your findings. Looking at the analysis for a more recent model's data would help me make a better decision for the paper.
> > >
> > > **A2:** We provide results rewritten using the newly released Gemini 3 for your reference:(the bolded parts are added content, questions generated using GPT, and asked using gpt-4o-mini)
> > > | Method           | 2020   | 2021   | 2022   | 2023   | 2024   |
> > > |------------------|--------|--------|--------|--------|--------|
> > > | Direct Ask       | 75.86% | 71.74% | 80.00% | 77.46% | 66.67% |
> > > | RAG              | 85.34% | 86.31% | 89.49% | 87.09% | 83.33% |
> > > | RAG (GPT)        | 85.63% | 88.96% | 87.18% | 87.09% | 84.58% |
> > > | RAG (Gem)        | 79.60% | 79.69% | 84.10% | 83.33% | 82.08% |
> > > | **RAG (Gem3)**       | 85.63% | 82.56% | 85.90% | 84.98% | 83.33% |
> > > | Full (Original)  | 95.98% | 96.03% | 95.64% | 96.01% | 95.83% |
> > > | Full (GPT)       | 95.40% | 96.03% | 95.64% | 94.84% | 95.83% |
> > > | Full (Gem)       | 87.36% | 88.08% | 88.97% | 87.09% | 88.75% |
> > > | **Full (Gem3)**      | 94.25% | 93.38% | 93.85% | 93.19% | 90.42% |
> > >
> > > Compared to the results rewritten with Gemini1.5, RAG (Gem3) and Full(Gem3) show significant improvements, almost matching the GPT rewrite and origin results, especially on Full. This indicates that as the model becomes more advanced, the risk of information loss may be mitigated.
> > > Thank you for your advice. We will further explore the use of more advanced models in future work to extend our experimental analysis. If you have any models you would suggest for testing, we would be happy to add them.
> > >
> > > ---
> > >
> > > **RQ3:** Please address the concerns I have raised regarding the representation of your experimental setup as it would strongly impact my decision for the submission
> > > I have also compiled some other cosmetic comments that would help improve the delivery of the paper.
> > >
> > > **A3:** If you have any further questions or concerns, please do not hesitate to point them out.
> > >
> > > ---
> > >
> > > **C1:** I don't quite understand what is fi\*(S) even after reading the text. I am also confused as to how you can determine if some text is affected by an LLM based on the formula. What is \\eta^{\\hat}(S) in Eq 1 supposed to capture, intuitively speaking and how does the interaction between different terms capture that?
> > >
> > > **A1:** Thank you for your question, and we apologize for not having explained it clearly earlier.
> > > Based on our previous definitions, we first have the following relation:
> > > \\begin{equation}
> > >         f^d\_{i}(S) \- f^\*\_{i}(S) \=  \\eta(S)f^\*\_{i}(S)\\hat{r}\_{i} \+ \\delta\_{i}(S)
> > > \\end{equation}
> > > where $\\delta\_{i}(S)$ is a noise term.
> > > Subsequently, the estimate of $\\eta(S)$ is derived via Ordinary Least Squares (OLS), which is Eq 1\.
> > >
> > > ---
> > >
> > > **C2:** Why is there no i anywhere in f(S2) or f(S1) in Eq 2?
> > >
> > > **A2:** Thanks for noting that. We are sorry for the confusion caused by our earlier expression. It should actually be $f\_i (S\_1)$ and $f\_i(S\_2)$.
> > >
> > > ---

---

> > > ### Comment · Reviewer_dwkh · 2025-12-28
> > >
> > > Thanks for adding some qualitative examples here -- they help me understand what's going on much clearly!

---

> > ### Comment · Reviewer_dwkh · 2025-12-28
> >
> > Thanks, you answers address most of my queries

---

> ### Author Response · Authors · 2025-12-16
> **Author Response 4 to Reviewer dwkh**
>
> **C3:** I don't quite understand the LLM Simulation setting, we have GPT-4o-mini and Gemini-1.5-Flash, but it is not clear from the main text what are they exactly simulating or benchmarked on. I took a quick look at the appendix but there's a lot going on there. It would really help if you could clarify the simulation setting more clearly in the main text.
>
> **A3:** (1) **Word Frequency:** We use *GPT-4o-mini* to revise the January 1, 2022, versions of Featured Articles, with the prompt “Revise the following sentences:”
> (2) **Linguistic Style:** We simulate the real Wikipedia with *GPT-4o-mini* and *Gemini-1.5-Flash*, then compare the changes before and after the process. Specifically, we instruct both models to revise Featured and Simple articles using the prompt mentioned above, and additionally use *GPT-4o-mini* to generate Wikipedia-style articles using the prompt “Generate a Wikipedia-style article titled {title} and return only the article body in plain text.”
> (3) **Machine Translation:** We use *GPT-4o-mini* to translate this English sentence into the remaining languages with the prompt “Translate the following text to {target language}”.
> (4) **RAG:** *GPT-4o-mini* and *Gemini-1.5-flash* are used to generate multiple-choice questions (MCQs) based on Wikinews articles. *GPT-4o-mini, GPT-3.5* are used to answer these questions under different settings.
>
> ---
>
> **C4:** The results section is also a bit hard to follow. In the sentence "Although we have plotted...from 2020 in these figures" do they mean the figures in the main text or the appendix? Following that, in "...the trends summarized in the table are based on..." which table does the highlighted "the" refer to? Table 1 caption does not clarify this either.
>
> **A4:** Thank you for pointing this out. The sentence “Although we have plotted the results from 2020 in these figures, the trends summarized in the table are based on the data in the LLM era, that is, after 2023” refers to the figures displaying metric trends over multiple years, which can be found both in the main text and the appendix. The adoption of LLMs became widespread around 2023, which means the impact of LLMs is supposed to take place after 2023\. Therefore, when we summarize the trends in Table 1, we specifically focus on the post-2023 period to capture trends of real Wikipedia pages. And then we compare them to LLM-driven trends to see if these two are consistent.
>
> ---
>
> **C5:** The term "page view" is a little ambiguous in this context as it can mean the number of views for a given wikipedia page (the intended meaning) or it could mean the rendered view of a Wikipedia page (which is how it reads like at the beginning).
>
> **A5:** We thank the reviewer for pointing out the ambiguity of the term “page view”. Wikipedia’s own definition is: “Page view statistics is a tool for Wikipedia pages which shows how many people have visited an article in a given time period.” To avoid misunderstandings, we have updated the wording in the Page View section. The revised text now explicitly states: “Page views (i.e., the number of times a Wikipedia page is accessed by users) …”
>
> ---
>
> **C6:** What does the "Full Content" setting mean in Figure 8? The input is the question+the wiki article that was used to construct those questions? Those setting names from the figure should be referenced in the main text.
>
> **A6:** You are correct. “Full Content” means that we query the models with the corresponding Wikinews articles included in the input. We have updated the Questioning Methods section to explicitly reference the prompts used for each method and to clearly explain the meaning of the setting names shown in Figure 8\.
>
> ---
>
> **C7:** Where do the chatGPT processed articles come into picture? It abruptly appears in the questioning methods paragraph and the exact protocol on how it was used is not clear.
> Results of the case study are really good.
>
> **A7:**  We have updated the Questioning Methods section to explicitly reference the prompts used for each method and to clearly explain the meaning of the setting names shown in Figure 8\.
>
> ---

---

> > ### Author Response · Authors · 2025-12-16
> > **Author Response 5 to Reviewer dwkh**
> >
> > **C8:** The authors have more than 1 page to make the main text more complete without changing the category of the submission, and I recommend utilizing it as I felt a lot of disconnections when reading the paper as some key details that would've helped me understand the paper better were absent.
> >
> > **A8:** We appreciate the suggestion and have used the available space to clarify and expand our experimental setup as follows:
> >
> > 1. **Extended time coverage:** We now include data spanning from 2018 to 2025\.
> > 2. **Direct Impact 1 – Word Frequency:** We have moved the explanation of word combination selection into the main text for better clarity.
> > 3. **Direct Impact 3 – Page Views:** To enhance generalizability beyond English Wikipedia, we now include page view analysis of Featured Articles in four major language editions: German (de), Spanish (es), English (en), and French (fr). And our conclusion is still consistent.
> > 4. **Indirect Impact 1 – Machine Translation:** We have revised the benchmark construction description for our machine translation experiments to improve clarity.
> > 5. **Indirect Impact 2 – RAG:** The Questioning Methods section has been updated to explicitly reference the prompts used for each method and to clarify the meaning of the setting names shown in Figure 8\.

---

### Review · Reviewer_SkDf · 2025-11-29

**Summary Of Contributions:**

This article quantitatively analyzes LLM's impact on Wikepedia, including first-order impact such as frequency, linguistics style, page view; and second-order impact, such as machine translation benchmarking and RAG that rely on Wikepedia. It's a comprehensive study.

It calls upon an important problem, the impact of LLM on human knowledge production. Specifically, it uses Wikipedia as a key venue of human knowledge production: it's originally written by humans, read by humans, and now used to train LLMs, and also presumably edited by LLMs. Had any impact on Wikipedia by LLMs, the impact can take years to be seen, given the 2nd-order impact may take time to be noticed. Such impact can be subtle, nontheless being impact, and consequence being serious.

The results are important contributions. Results on both 1st-order and 2nd-order impact are notable and largely convincing. The authors stated LLM impact on Wikipedia is "approximately 1%-2% in certain categories"(figure 3), it notes that the linguistic style changes in Wikepedia highly correlated to the linguistic style changes in LLM simulations (table 1). The 2nd degree impact might be some of the earlies efforts on this rather neglected problem, suggesting that LLM's impact on Wikepedia does not stop at Wikipedia.

It identifies a set of methodologies to study LLM impact on human knowledge: word frequency compared to baseline frequency before GPT, linguistic style change that correlates to LLM-simulated linguistic changes, constructing benchmark and RAG based on LLM-revised wikipedia to understand the impact. Overall they are rather comprehensive.

**Audience:**

Yes

**Audience Explanation:**

Yes. I believe the work on LLM impact on human knowledge production and transmission require attention from ML community, as LLM, different from other more conventional ML technologies are widely adopted in many research, educational, and information-processing tasks. Such impact is poorly studied, and it requires long-term monitor, investigation, and even causal analysis.

**Broader Impact Concerns:**

No ethical concerns applied. This is an empirical analysis of LLM impact on Wikipedia.

**Claims And Evidence:**

Yes

**Claims Explanation:**

Overall claims are overall supported by presented methdology and evidence, and they are mostly accurate and convincing.

- The 1st-degree impact is overall robust. "approximately 1%-2% in certain categories" are calculated across catagories and are adopted from previously established methodology from (Geng et al., 2025).
- The linguistic style changes largely correlates to the LLM-based simulations that authors ran.
- The page view presents mixed results, and there can be reasons beyond LLM that explain the drop. But this is not the main results of this paper.
- The 2nd-degree impact such as that on MT benchmark and RAG seem exploratory, as the authors construct (rather toy) benchmars and RAGs based on LLM-rewritten content, giving readers a flavor what might happen in such set up. The resuts are mixed, and the methologyies seem more for demonstration rather than rigorious study. I regarded them as some early effort to investigate 2nd-order problems. And the authors present some relevant findings.

**Requested Changes:**

There are some methodological problems to be noted, and changes might be needed:

- (crucial) In the assessment of word frequency, the data was taken as comparison from 2020-2021 on the same articles. There are two probems here (i) 2020-2021 seems a very unsuaul year in human history because of covid-19. An example of anomaly:  and notably there are far more studies about covid-19 relevant topics from other years; (ii) the comparison here is between "word frequency of Wikipedia *with* LLM-editing" and "word frequency of Wikipedia *without* LLM-editing", but the authors didn't investigate whether word frequency would usually change, too. The question here should be "whether the word frequency change under LLM is rarther unsusal compared to baseline change? This is not investigated by the authors.
- As authors noted themselves "Second, the lack of field experiments limits our insights into the actual machine-in-the-loop editing processes behind Wikipedia article creation." Indeed, ineractive dynamic should be far more complex than what they did with simulations. And understanding that interactive dynamic might be hard. But can authors estimate whether their results are at upper or lower bound of the impact? This is a comment on linguistic style subsection.
- (crucial) The 2nd-order impact overall is unrealitic. If your own estimation is 1-2% impact (adimittedly, what does 1-2% mean is unclear. It's linguistic frequency difference, but I am not sure whether we can intuitively think that in average 1~2% of each article is re-written by LLM), why do you try benchmark/RAG based on 100% rewritten by LLMs? As a natural result, the impact is inflated. Wouldn't it be more convincing to try lighter LLM revision? As you argue, the LLM impact grows over time, and in some catagories the impact exceeds 1-2%. So how about trying LLM revision in the bar park of 1-2% but slightly higher? This becomes a straw-man argument.
- Gemini 1.5 Flash was released early 2024 and it's a rather low-capability model. I won't be too suprised that RAG / MT benchmark based on it would experience accuracy drop. But what about newer models? If RAG based on Gemini 3-revised Wikepedia can achieve human performance, would we claim this is a lesser concern over time? Also, if GPT-4o-mini does not compromise RAG performance as well, would this be an artifact with Gemini 1.5 Flash?

---

> ### Author Response · Authors · 2025-12-16
> **Author Response 1 to Reviewer SkDf**
>
> Thank you for your questions and suggestions. We hope our response addresses your concerns.
>
> **RQ1:** (crucial) In the assessment of word frequency, the data was taken as comparison from 2020-2021 on the same articles. There are two problems here (i) 2020-2021 seems a very unusual year in human history because of covid-19. An example of anomaly: and notably there are far more studies about covid-19 relevant topics from other years; (ii) the comparison here is between "word frequency of Wikipedia with LLM-editing" and "word frequency of Wikipedia without LLM-editing", but the authors didn't investigate whether word frequency would usually change, too. The question here should be "whether the word frequency change under LLM is rather unusual compared to baseline change? This is not investigated by the authors.
>
> **A1:** (i) Thank you for pointing this out, and our method actually intentionally avoids it. A key step in our study is selecting appropriate word combinations for estimating the impact of LLMs. We apply a threshold on f\* to ensure that the selected words appear frequently across the corpus, and we further set a threshold on r to guarantee that these words exhibit a substantial frequency change after being processed by the LLM. Topic-specific words (e.g., “pandemic,” “covid”) were indeed unusually common during 2020 and 2021, but such terms typically do not undergo much frequency changes after LLM processing. So they are not likely to be chosen for impact estimation. In contrast, the words identified by our selection procedure tend to be general-purpose words. As shown in our appendix, examples include “within,” “caused,” “ended,” “then,” “subsequent,” etc. These examples illustrate that our analysis is not driven by topic-related words but focus on words that can be used in different scenarios.
>
> (ii) We have expanded our dataset to include Wikipedia page versions from 2018 and 2019\. To address this concern, we first focus on the 2018–2022 period and fit linear models to characterize the natural evolution of Wikipedia articles before the widespread adoption of LLMs. We then extrapolate this pre-LLM trend into the post-2022 period and measure deviations relative to this baseline.
>
> As shown in Figure 3 of the revised manuscript, the estimated impact for several Wikipedia categories exhibits only stable fluctuations before 2022, followed by a substantial increase after 2022\. This pattern indicates that the post-2022 shifts are not simply continuations of earlier trends but represent deviations from a historically stable baseline. By explicitly constructing the baseline using pre-LLM data, our approach helps separate long-term evolution from the causal effect of LLMs.
>
> ---
>
> **RQ2:** As authors noted themselves "Second, the lack of field experiments limits our insights into the actual machine-in-the-loop editing processes behind Wikipedia article creation." Indeed, interactive dynamics should be far more complex than what they did with simulations. And understanding that interactive dynamic might be hard. But can authors estimate whether their results are at the upper or lower bound of the impact? This is a comment on the linguistic style subsection.
>
> **A2:** Our design intentionally applies full LLM revision to get LLM preference and provide a measurable estimate of LLM influence. Therefore, our simulation should be interpreted as an upper bound on the potential impact of LLMs, under certain assumptions (e.g., using the given LLMs and prompt). In real editing workflows, human editors typically adopt only a subset of LLM-suggested changes rather than accepting a complete rewrite, meaning the actual influence on Wikipedia is likely lower.
>
> ---

---

> > ### Comment · Reviewer_SkDf · 2026-01-14
> > **Both A1 and A2 clarified my concerns.**
> >
> > Both A1 and A2 make sense to me. Thanks for additional detail!

---

> ### Author Response · Authors · 2025-12-16
> **Author Response 2 to Reviewer SkDf**
>
> **RQ3:** (crucial) The 2nd-order impact overall is unrealistic. If your own estimation is 1-2% impact (admittedly, what does 1-2% mean is unclear. It's a linguistic frequency difference, but I am not sure whether we can intuitively think that on average 1\~2% of each article is re-written by LLM), why do you try benchmark/RAG based on 100% rewritten by LLMs? As a natural result, the impact is inflated. Wouldn't it be more convincing to try a lighter LLM revision? As you argue, the LLM impact grows over time, and in some categories the impact exceeds 1-2%. So how about trying LLM revision in the bar park of 1-2% but slightly higher? This becomes a straw-man argument.
>
> **A3:** For machine translation, benchmark construction involves translating a single sentence multiple times. The unit of evaluation is already minimal, and selectively rewriting fragments within a single sentence would no longer correspond to a realistic translation task.
>
> For RAG, partial revision is also difficult to do. During question generation, we do not know in advance which specific sentence(s) will contain the answer for that question. Applying partial revision would introduce arbitrary biases about which parts of the document to alter, while simply prompting LLMs for revision ensures that every sentence has an equal probability of being modified. This makes the experimental setup clearer and the results easier to interpret.
>
> More importantly, when we prompt the model to “revise the following sentences,” this does not mean a complete rewrite. It typically results in moderate edits that preserve some of the original words/style/content. The prompt has actually been widely used in many research about LLM-generated text, and it is proved to be effective.
>
> ---

---

> ### Author Response · Authors · 2025-12-16
> **Author Response 3 to Reviewer SkDf**
>
> **RQ4:** Gemini 1.5 Flash was released early 2024 and it's a rather low-capability model. I won't be too surprised that the RAG / MT benchmark based on it would experience an accuracy drop. But what about newer models? If RAG based on Gemini 3-revised Wikipedia can achieve human performance, would we claim this is a lesser concern over time? Also, if GPT-4o-mini does not compromise RAG performance as well, would this be an artifact with Gemini 1.5 Flash?
>
> **A4:**  We generally agree with your point of view.  First, comparing RAG (Origin, GPT, Gemini) and Full (Origin, GPT, Gemini), we find that the accuracy of the GPT column is not significantly different from the Origin column, while the accuracy of the Gemini column drops significantly. A reasonable explanation is that some key information was omitted during the Gemini rewrite of wikinews, causing LLM to be unable to find the materials needed for the corresponding questions (because the questions, prompts, and test LLMs have not changed; only the reference knowledge has changed). We think that LLMs have the potential for information loss when processing text, but some models (like Gemini 1.5 Flash) are more susceptible to this risk than others.
>
> As for whether this is a concern over time, I believe that with the replacement of advanced models, this risk may indeed be reduced (for example, the GPT column does not show a significant decrease compared to the Origin column). However, the impact of LLM on wikinews web pages is permanent. If information is lost during a revision, it may be hard to recover in the next revision using an advanced model. For those works that used this information for downstream tasks (such as RAG), its core components were permanently lost during that version revision, and no amount of subsequent algorithm optimization or model training can compensate for it. Therefore, I believe we still need to remain concerned.
>
> In addition, we have also provided results rewritten using the newly released Gemini3 for your reference:(the bolded parts are added content, questions generated using GPT, and asked using gpt-4o-mini)
> | Method           | 2020   | 2021   | 2022   | 2023   | 2024   |
> |------------------|--------|--------|--------|--------|--------|
> | Direct Ask       | 75.86% | 71.74% | 80.00% | 77.46% | 66.67% |
> | RAG              | 85.34% | 86.31% | 89.49% | 87.09% | 83.33% |
> | RAG (GPT)        | 85.63% | 88.96% | 87.18% | 87.09% | 84.58% |
> | RAG (Gem)        | 79.60% | 79.69% | 84.10% | 83.33% | 82.08% |
> | **RAG (Gem3)**       | 85.63% | 82.56% | 85.90% | 84.98% | 83.33% |
> | Full (Original)  | 95.98% | 96.03% | 95.64% | 96.01% | 95.83% |
> | Full (GPT)       | 95.40% | 96.03% | 95.64% | 94.84% | 95.83% |
> | Full (Gem)       | 87.36% | 88.08% | 88.97% | 87.09% | 88.75% |
> | **Full (Gem3)**      | 94.25% | 93.38% | 93.85% | 93.19% | 90.42% |
>
> As you can see, compared to the results rewritten with Gemini1.5, RAG (Gem3) and Full (Gem3) show significant improvements, almost matching the GPT rewrite and origin results, especially on Full. This indicates that as the model becomes more advanced, the risk of information loss may be mitigated, which is consistent with your thinking.
>
> Thank you for your advice. We will also further explore the use of more advanced models in future work to extend our experimental analysis.

---

### Review · Reviewer_Y14v · 2025-11-30

**Summary Of Contributions:**

The authors present an empirical analysis of how LLMs are already influencing Wikipedia and how such influence may affect downstream NLP tasks, especially MT benchmarks and RAG.

The authors collect Wikipedia snapshots (2020–2025), Featured Articles, Simple English articles, and Wikinews articles. They conduct several direct and indirect impact analyses. Their direct analyses include estimating LLM influence through word-frequency patterns based on simulated GPT-edited pages. They also track linguistic-style trends at the word, sentence, and paragraph levels, finding that many recent changes in Wikipedia increasingly align with typical LLM writing tendencies. Beyond these surface effects, the authors demonstrate indirect effects: GPT-translated variants of the FLORES raise model scores while LLM-rewritten Wikinews articles reduce factual retrieval accuracy in RAG systems due to information fusion, omission, etc.

Strengths:
- The paper is clearly written and easy to follow.
- The authors perform a multi-dimensional analysis (style, pageviews, MT, RAG) which is helpful to understand the problem.
- The authors provide a strong basis with real Wikipedia data and extensive simulations.
- The authors utilised a number of techniques in direct analysis to measure LLM impact. The trends provided are quite useful.
- Declining accuracy for recent trends: I really liked this subsection how LLMs are really good when it comes to old data and do quite bad with new data.

Weakness:
- Many of the paper’s findings are correlational. The analyses rely on naturally occurring Wikipedia edits and pageview trends, which can be influenced by many factors besides LLMs. For example, shifting editor behavior, policy changes, etc. Moreover, the LLM rewriting process which authors talk about may not capture actual human-in-the-loop Wikipedia editing behaviour.
 - Adding to point above, one way of separating causality with correlation is to check if data from years before 2020 also follow certain trend when compared to post 2020?
- The downstream effects on MT and RAG are demonstrated through controlled simulations rather than evidence of actual contamination. Therefore, the paper suggests possible mechanisms but does not establish definitive causal relationships.
- The source text for FLORES200 came from Wikipedia but to ensure accuracy and fidelity, the translations were manually produced by professional, native-speaking translators. Picking up samples directly from Wikipedia for benchmarks is problematic because of data contamination-- LLMs have swallowed Wikipedia multiple times by now, existing LLMs will do really well on those benchmarks and as a result those benchmarks will become outdated the minute it comes out.
- The LLMs produce “translationese” when prompted to translate from one language to another. As a result, other models score really high on the GPT benchmark whereas Wikipedia follows human-in-the-loop editing behavior in which contributors can replace translationese with culture-specific terms.
- The authors mention a bunch of linguistic style trends in Table 1 but offer no definition for bottom 6 and how they are relevant to their objective.

**Audience:**

Yes

**Audience Explanation:**

The paper talks about major NLP concerns: dataset contamination, benchmark reliability, LLM-driven language drift, and long-term risks for knowledge bases. Moreover, since Wikipedia data is involved in training and evaluating modern models, the audience will find these results relevant. These issues matter directly to:
- NLP researchers using Wikipedia-based corpora.
- MT researchers relying on FLORES-like benchmarks.
- People working on RAG, knowledge-retrieval and LLM influence on human-generated content.

**Broader Impact Concerns:**

- Wikipedia might need new policies for detecting or managing LLM-assisted edits to maintain the original linguistic styles. The paper should more explicitly discuss such mechanisms as well.
- The authors showed how LLM contaminated articles will impact benchmarking but didn't how will it affect the models that are trained on such data. Will they continue doing normal or their performance will be impacted too?

**Claims And Evidence:**

Yes

**Claims Explanation:**

The paper provides clear evidence for each major claim:
- Word-frequency impact is quantified using an estimator applied to multiple word subsets.
- Linguistic-style alignment is demonstrated with clear metrics and LLM simulations, showing consistent trends across stylistic categories.
- MT inflation effects are supported with controlled experiments on 11 languages and multiple metrics, showing score increases.
- RAG degradation is demonstrated with MCQs and multiple LLMs, including detailed case studies of where and why LLM-edited content fails.

**Requested Changes:**

- My main concern is that the paper occasionally implies LLMs “caused” stylistic changes or page view declines, though evidence is correlational. State limits clearly and don’t imply cause-and-effect without evidence. It will be also great if authors can show the same trend for a language other than English to corroborate their claims.
- Real Wikipedia edits may involve partial edits, human-machine interactions with different LLM quality. I believe that revising the whole article with the GPT approach may overestimate the stylistic impact. The changes should be a bit subtle just like how changes happen in Wikipedia.

---

> ### Author Response · Authors · 2025-12-16
> **Author Response 1 to Reviewer Y14v**
>
> Thank you for your questions and suggestions. We hope our response addresses your concerns.
>
> **W1:** Many of the paper’s findings are correlational. The analyses rely on naturally occurring Wikipedia edits and pageview trends, which can be influenced by many factors besides LLMs. For example, shifting editor behavior, policy changes, etc. Moreover, the LLM rewriting process which authors talk about may not capture actual human-in-the-loop Wikipedia editing behaviour.
>
> **A1:** The first part of our results (3 direct impacts) show correlation, while the second part (2 indirect impacts) can be considered causal. In addition, related work generally investigates correlation rather than causality. We agree that our simulations cannot fully capture the real-world editing process, which might also be an impossible task to accomplish. However, our goal is not to prove that LLMs are the only cause of observed changes, but to show they are a sufficient cause. Given the widespread use of LLMs, partial causal effects are enough to pose significant risks.
>
> ---
>
> **W2:** Adding to the point above, one way of separating causality with correlation is to check if data from years before 2020 also follow a certain trend when compared to post 2020?
>
> **A2:** Thank you for your suggestion. We have expanded our dataset to include Wikipedia page versions from 2018 and 2019\. To address this concern, we first focus on the 2018–2022 period and fit linear models to characterize the natural evolution of Wikipedia articles before the widespread adoption of LLMs. We then extrapolate this pre-LLM trend into the post-2022 period and measure deviations relative to this baseline.
>
> As shown in Figure 3 of the new version, the estimated impact for several Wikipedia categories exhibits only stable fluctuations before 2022, followed by a substantial increase after 2022\. This pattern indicates that the post-2022 shifts are not simply continuations of earlier trends but represent deviations from a historically stable baseline. By explicitly constructing the baseline using pre-LLM data, our approach helps separate long-term evolution from the causal effect of LLMs.
>
> ---
>
> **W3:** The downstream effects on MT and RAG are demonstrated through controlled simulations rather than evidence of actual contamination. Therefore, the paper suggests possible mechanisms but does not establish definitive causal relationships.
>
> **A3:** Since our analysis on the direct impacts already shows that LLMs might have begun to influence Wikipedia, the goal of our indirect-impact analysis is not to determine whether existing Wikipedia-based downstream datasets have been contaminated. Instead, what we try to investigate is how NLP tasks that rely on a potentially contaminated Wikipedia corpus would behave under such LLM influence. In theory and simulations, such causality exists, but in practice, the situation is more complex.
>
> ---
>
> **W4:** The source text for FLORES200 came from Wikipedia but to ensure accuracy and fidelity, the translations were manually produced by professional, native-speaking translators. Picking up samples directly from Wikipedia for benchmarks is problematic because of data contamination-- LLMs have swallowed Wikipedia multiple times by now, existing LLMs will do really well on those benchmarks and as a result those benchmarks will become outdated the minute it comes out.
>
> **A4:** We agree that benchmarks are built from real-world data and may be included in LLM training corpora. But we are exploring the potential impact on the evaluation of machine translation models if LLMs influence the benchmarks for machine translation. Hence, this part does not have the problems you are concerned with.
>
> ---
>
> **W5:** The LLMs produce “translationese” when prompted to translate from one language to another. As a result, other models score really high on the GPT benchmark whereas Wikipedia follows human-in-the-loop editing behavior in which contributors can replace “translationese” with culture-specific terms.
>
> **A5:** We totally agree. Current machine translation models might struggle to generate cultural-specific expressions, and may instead produce “translationese” resembling LLM-influenced benchmarks. This might partially explain the observed increase in model scores. We actually do not consider this hypothesis as a weakness as it further highlights the potential risks of relying on benchmarks that may already be influenced by LLM-generated content.
>
> ---

---

> > ### Author Response · Authors · 2025-12-16
> > **Author Response 2 to Reviewer Y14v**
> >
> > **W6:** The authors mention a bunch of linguistic style trends in Table 1 but offer no definition for bottom 6 and how they are relevant to their objective.
> >
> > **A6:** The six paragraph-level metrics reflect the readability of Wikipedia articles. We have added the detailed definitions of them in the appendix “A3.3 Paragraph Level” in our revised version. As noted in our “Paragraph Level” section, readability is essential for Wikipedia’s educational mission (Johnson et al., 2024b). Our goal is to examine whether the readability of Wikipedia pages has shifted in the era of LLMs, which makes these metrics directly relevant to our analysis.
> >
> > ---
> >
> > **Requested Changes:**
> >
> > **RQ1:** My main concern is that the paper occasionally implies LLMs “caused” stylistic changes or page view declines, though evidence is correlational. State limits clearly and don’t imply cause-and-effect without evidence. It will be also great if authors can show the same trend for a language other than English to corroborate their claims.
> >
> > **A1:** Thanks for your suggestion. We have changed some expressions. To generalize our findings beyond English Wikipedia, we further analyze the page views of Featured Articles in four major language editions, German (de), Spanish (es), English (en), and French (fr) Wikipedia. Similar to the English case, these languages also exhibit a decline beginning in 2024, with the drop being especially sharp in the Spanish edition.
> >
> > ---
> >
> > **RQ2:** Real Wikipedia edits may involve partial edits, human-machine interactions with different LLM quality. I believe that revising the whole article with the GPT approach may overestimate the stylistic impact. The changes should be a bit subtle just like how changes happen in Wikipedia.
> >
> > **A2:** Our design intentionally applies full LLM revision to get LLM preference and provide a measurable estimate of LLM influence. Therefore, our simulation could be interpreted as an upper bound on the potential impact of LLMs under some assumptions. In real editing workflows, human editors typically adopt only a subset of LLM-suggested changes rather than accepting a complete rewrite, meaning the actual influence on Wikipedia is likely lower.
> >
> > ---
> >
> > **Broader Impact Concerns:**
> >
> > **C1:** Wikipedia might need new policies for detecting or managing LLM-assisted edits to maintain the original linguistic styles. The paper should more explicitly discuss such mechanisms as well.
> >
> > **C2:** The authors showed how LLM contaminated articles will impact benchmarking but didn't how it will affect the models that are trained on such data. Will they continue doing normal or their performance will be impacted too?
> >
> > **A1+A2:** We added the following sentences to our “Estimation of LLM Impact.” in “related work” section:
> > Moreover, The emergence of LLM-assisted edits on Wikipedia has raised concerns about preserving the encyclopedia’s consistent style and editorial standards (Ashkinaze et al., 2024). Contamination of its articles with LLM-generated text can create harmful feedback loops in model training (Shumailov et al., 2023).

---

### Decision · Action_Editor_WwHg · 2026-01-20

**Recommendation:** Accept as is

**Additional Comments:**

The reviewers gave helpful suggestions for making the paper more clear, some of which have already been incorporated. Please be sure to implement these suggestions.

**Audience:**

Yes

**Audience Explanation:**

At least some individuals will find this paper's findings interesting. This will likely be the pretraining and potentially post-training community because Wikipedia is frequently used for training data. There are also two case studies on how machine translation and RAG are impacted as well, so researchers in these fields may also be interested. All reviewers agree.

**Claims And Evidence:**

Yes

**Claims Explanation:**

This paper's primary claims are that the bidirectional relationship between LLMs and Wikipedia may lead to negative impacts. The findings are that Wikipedia has been directly impacted by LLMs, which subsequently indirectly impacts RAG and machine translation (MT) tasks. To support the claims of Wikipedia being impacted, the paper uses concrete experiments that look at changes in word frequency, linguistic style and page views. For the indirect impacts, there are two case studies that show that MT scores artificially inflate yet RAG suffers.

2/3 reviewers initially agreed that the claims are well supported in the paper. The third reviewer agreed after the discussion period with the authors. I've read these reviews and paper and agree.